# Repressing *Ago2* mRNA translation by Trim71 maintains pluripotency through inhibiting let-7 microRNAs

**Qiuying Liu[1], Xiaoli Chen[2], Mariah K Novak[1], Shaojie Zhang[2], Wenqian Hu[1]\***

[1]Department of Biochemistry and Molecular Biology, Mayo Clinic, Rochester, United States; [2]Department of Computer Science, University of Central Florida, Orlando, United States

**Abstract** The regulation of stem cell fate is poorly understood. Genetic studies in *Caenorhabditis elegans* lead to the hypothesis that a conserved cytoplasmic double-negative feedback loop consisting of the RNA-binding protein Trim71 and the let-7 microRNA controls the pluripotency and differentiation of stem cells. Although let-7-microRNA-mediated inhibition of Trim71 promotes differentiation, whether and how Trim71 regulates pluripotency and inhibits the let-7 microRNA are still unknown. Here, we show that Trim71 represses *Ago2* mRNA translation in mouse embryonic stem cells. Blocking this repression leads to a specific post-transcriptional increase of mature let-7 microRNAs, resulting in let-7-dependent stemness defects and accelerated differentiation in the stem cells. These results not only support the Trim71-let-7-microRNA bi-stable switch model in controlling stem cell fate, but also reveal that repressing the conserved pro-differentiation let-7 microRNAs at the mature microRNA level by Ago2 availability is critical to maintaining pluripotency.

## Introduction

The switch between pluripotency and differentiation in embryonic stem cells (ESCs) remains incompletely understood. Although nuclear events controlling stemness are becoming increasingly clear, how cytoplasmic pathways of gene expression regulate ESCs' fates between pluripotency and differentiation are still poorly understood (*Ye and Blelloch, 2014*).

Genetic studies in *C. elegans* led to the postulation that a conserved cytoplasmic bi-stable switch controls the pluripotency and differentiation of stem cells (*Ecsedi and Grosshans, 2013*). This switch is proposed to involve reciprocal negative regulation between the conserved pro-differentiation let-7 microRNA (miRNA) and Trim71 (Lin41 in *C. elegans*), a conserved and ESC-specific RNA-binding protein (RBP). The following observations support this model. First, the let-7 miRNA negatively correlates with Trim71 during stem cell differentiation: the let-7 miRNA level increases, while Trim71 decreases during differentiation. Second, Trim71 is a conserved target of the let-7 miRNA, and repressing Trim71 by let-7 promotes stem cell differentiation (*Aeschimann et al., 2019*; *Ecsedi et al., 2015*; *Grishok et al., 2001*; *Roush and Slack, 2008*). Third, inhibiting Trim71 suppresses developmental defects caused by mutations in the core components of the miRNA pathway in *C. elegans* (*Büssing et al., 2010*; *Grishok et al., 2001*), suggesting that Trim71 may negatively regulate the miRNA pathway. Thus, it is hypothesized that let-7 miRNA and Trim71 reciprocally repress each other. This double-negative feedback loop forms a molecular bi-stable switch, in which stem-cell differentiation is controlled by the let-7-miRNA-mediated inhibition of Trim71 and pluripotency is controlled by the hypothetical Trim71-mediated inhibition of the let-7 miRNA (*Ecsedi and Grosshans, 2013*). Due to the conservation of let-7 miRNA, Trim71, and the let-7-mediated inhibition of Trim71, the cytoplasmic bi-stable switch controlling stem cell fate is thought to be conserved

**\*For correspondence:**
hu.wenqian@mayo.edu

**Competing interests:** The authors declare that no competing interests exist.

in animals. A lingering question in this bi-stable switch model, however, is whether and how Trim71 inhibits the let-7 miRNA and regulates pluripotency in stem cells.

Trim71 was proposed to interact with and ubiquitylate Ago2, a critical component of the miRNA pathway, resulting in Ago2 degradation in mammalian cells (*Rybak et al., 2009*). Although the functional significance of this interaction to stem cell biology was not examined, this observation seemed to support the bi-stable switch model. Later studies, however, indicated that the Trim71-Ago2 interaction is RNA dependent (*Chang et al., 2012*; *Loedige et al., 2013*), and the proposed Trim71-mediated Ago2 degradation is absent in vivo (*Chen et al., 2012*; *Welte et al., 2019*). Thus, it is unclear how Trim71 modulates the let-7 miRNA. In terms of biological functions, Trim71 knockout mice are embryonic lethal (*Cuevas et al., 2015*), while Trim71 knockout mouse ESCs (mESCs) have no proliferation or stemness defects (*Chang et al., 2012*; *Mitschka et al., 2015*; *Welte et al., 2019*; *Worringer et al., 2014*), indicating an enigmatic role of Trim71 in stem cell biology. Collectively, these results highlight the hypothetical status of Trim71's function and mechanisms in the bi-stable switch model and beg for investigations on how Trim71 regulates the let-7 miRNAs and whether this regulation plays a role in controlling pluripotency in stem cells.

Here, we show that Trim71 maintains pluripotency through inhibiting the let-7 miRNAs. We identified the transcriptome-wide targets of Trim71 in mESCs and determined that Trim71 binds and represses *Ago2* mRNA translation. Specific disruption of this repression leads to an elevated Ago2 level, which results in a specific post-transcriptional increase of the mature let-7 miRNAs, decreased stemness, and accelerated differentiation in mESCs. These stem cell defects are dependent on the let-7 miRNAs, as specific inhibition of the let-7 miRNAs abolishes the stemness defects caused by the loss of Trim71-mediated repression of *Ago2* mRNA translation in mESCs. Collectively, these results provide direct support for the cytoplasmic bi-stable switch model of stem cell fate decision. Moreover, this study reveals that repressing the conserved pro-differentiation let-7 microRNAs at the mature miRNA level by Ago2 availability is critical to maintaining pluripotency.

## Results

### Transcriptome-wide identification of Trim71's target mRNAs in mESCs

To study Trim71's function in mESCs, we created bi-allelic FLAG-tagged Trim71 in mESCs. Using CRISPR/Cas9-mediated genomic editing, we inserted a FLAG-tag at the N-terminus of Trim71 (*Figure 1A*) and identified bi-allelic FLAG-tag knock-in mESC clones (*Figure 1B*). The knock-in sequence changes neither Trim71's native promoter nor the 3'UTR (untranslated region), where transcriptional and post-transcriptional regulations mainly occur, respectively, and the FLAG-Trim71 is expressed at the endogenous level (*Figure 1—figure supplement 1A*). Moreover, the FLAG-Trim71 mESC is phenotypically identical to the wild type (WT) mESC: they have similar morphology, growth rates, self-renewal abilities, and express similar levels of core pluripotency transcription factors (*Figure 1—figure supplement 1B–F*). Thus, we refer to the FLAG-Trim71 mESCs as the WT mESCs.

The FLAG-tag facilitates unambiguous detection and efficient isolation of the endogenous Trim71 in mESCs. Using an anti-FLAG monoclonal antibody, we could specifically detect Trim71 in the FLAG-Trim71 mESCs (*Figure 1C*). Moreover, most Trim71 could be immunoprecipitated (IP) from the FLAG-Trim71 mESCs lysate via the anti-FLAG antibody (*Figure 1D*). This IP is specific because: (a) in the IP using IgG, Trim71 remained in the supernatant; and (b) when the IP was performed in the control mESC without the FLAG-tag, the IP sample generated little signal (*Figure 1D*).

To determine whether Trim71 regulates mESCs, we identified transcriptome-wide targets of Trim71 in mESCs using cross-linking immunoprecipitation and sequencing (CLIP-seq) (*Figure 1E*; *Darnell, 2010*). This method not only revealed which mRNAs Trim71 binds but also identified the binding sites on those mRNAs. Trim71-binding sites are mainly located in the introns and 3'UTRs of the target mRNAs (*Figure 1F*; *Supplementary file 1*). Sequence analysis identified an over-represented stem-loop structure, but no enriched primary sequence motifs, in the Trim71-binding sites compared to randomized sequences (*Figure 1G*). This observation suggests that Trim71 recognizes RNA secondary structures, but not a primary sequence, which is consistent with recent in vitro and in vivo studies on Trim71:RNA interactions (*Kumari et al., 2018*; *Welte et al., 2019*). *Cdkn1a* mRNA (*Figure 1H*), a validated Trim71 target (*Chang et al., 2012*), is among the identified

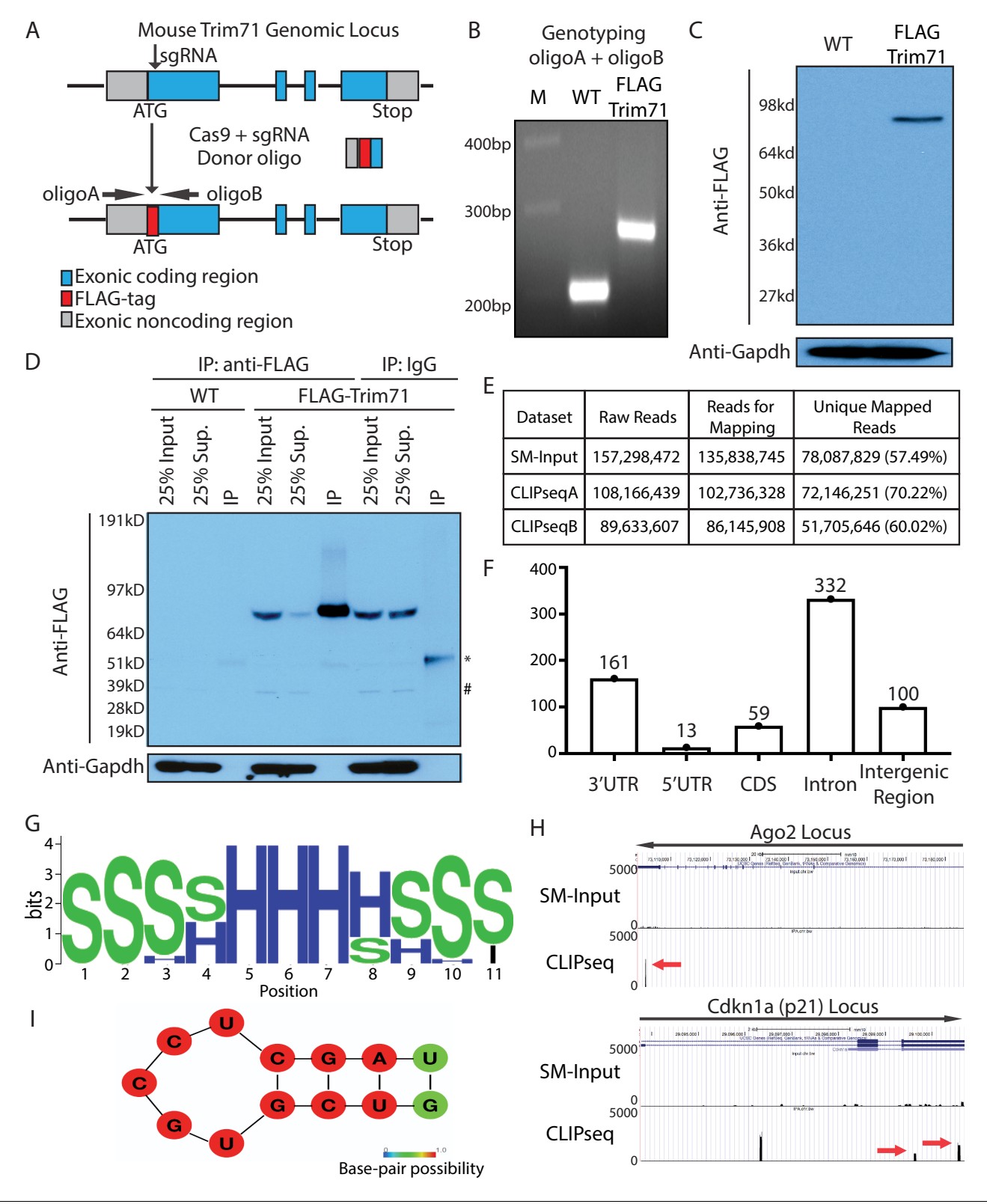

**Figure 1.** Transcriptome-wide identification of Trim71 target mRNAs in mouse embryonic stem cells (mESCs). (**A**) Workflow for knock-in the FLAG-tag to the endogenous Trim71 locus in mESCs. (**B**) Genotyping of the FLAG-Trim71 mESCs using the two primers in (**A**). (**C**) Specific detection of the endogenous Trim71 via the FLAG-tag. Western blotting in the WT and the FLAG-Trim71 mESCs using an anti-FLAG monoclonal antibody. (**D**) Efficient and specific isolation of the endogenous Trim71. An anti-FLAG monoclonal antibody and mouse IgG were used to immunoprecipitate (IP) the

*Figure 1 continued on next page*

*Figure 1 continued*

endogenous Trim71 from the lysates of the WT and the FLAG-Trim71 mESCs. The inputs, supernatants (Sup.), and IP samples were subject to SDS-PAGE and western blotting using the indicated antibodies. * IgG heavy chain; # a non-specific band. (E) A table summarizing the number of reads from the Trim71 CLIP-seq experiments (F) Distribution of Trim71 binding regions in the mouse genome. (G) RNA secondary structures over-represented in the Trim71 binding regions within the 3'UTRs of mRNAs. (H) UCSC genome browser snapshots for the two Trim71 target mRNAs. The red arrows indicate the Trim71 binding regions in 3'UTRs. (I) Predicted RNA secondary structure in the Trim71 binding region in *Ago2* mRNA's 3'UTR.

The online version of this article includes the following figure supplement(s) for figure 1:

**Figure supplement 1.** The FLAG-Trim71 mouse embryonic stem cells (mESCs) are phenotypically indistinguishable from the WT mESCs.

mRNAs with Trim71-binding sites in the 3'UTR. This observation argued for the validity of the 3'UTR Trim71-binding sites we identified.

In this study, we focused on the Trim71:*Ago2*–mRNA interaction because: (a) the *Ago2*'s 3'UTR contains only one Trim71-binding site with the predicted stem-loop structure (*Figure 1H and I*); (b) this binding site is also present in a recent study on identifying transcriptomic-wide targets of Trim71 *Welte et al., 2019*; (c) genetic studies in *C. elegans* suggest that Trim71 has links to the miRNA pathway (*Ecsedi and Grosshans, 2013*), in which Ago2 is a key component.

## Specific inhibition of the Trim71's binding on *Ago2* mRNA

Previous studies indicated that knocking out/down Trim71 had no impact on Ago2 (*Chang et al., 2012*; *Welte et al., 2019*), which we recapitulated in our mESCs (*Figure 2—figure supplement 1A*). One caveat of this loss-of-function approach, however, is that hundreds of Trim71:mRNA interactions and potential Trim71-mediated protein interactions are lost in Trim71 knockout cells, making it difficult to evaluate the functional significance of a specific Trim71:mRNA interaction (e.g., Trim71:*Ago2*–mRNA interaction in this study).

To specifically investigate the function of the Trim71:*Ago2*–mRNA interaction, we deleted the Trim71-binding region (115 bp), defined from the CLIP-seq (*Figure 1H*), in the 3'-UTR of *Ago2* mRNA using genome editing. We identified two independent mESC clones with bi-allelic deletions, which we named CLIPΔ clones (*Figure 2A*, *Figure 2—figure supplement 1B*). RNA-seq revealed similar reads intensity and distribution across *Ago2* 3'UTR except the deleted Trim71-binding region among the WT and the two CLIPΔ clones (*Figure 2—figure supplement 1C*), indicating no large DNA fragment deletion caused by the genome editing in the target region. CLIP-qRT-PCR indicated that Trim71 in the CLIPΔ mESCs does not bind *Ago2* mRNA, but still specifically interacts with other target mRNAs, such as *Cdkn1a* mRNA (*Figure 2—figure supplement 1D and E*). Thus, the CLIPΔ cells enabled us to specifically examine the function of the Trim71:*Ago2*–mRNA interaction in mESCs.

## Trim71 represses *Ago2* mRNA translation in mESCs

Multiple lines of evidence indicated that Trim71 represses *Ago2* mRNA translation in mESCs.

First, Ago2 protein level increased approximately twofold without an increase of the mRNA in two independent CLIPΔ mESC clones compared to WT mESCs (*Figure 2B and C*). In the *Trim71* knockout (KO) genetic background, however, the CLIPΔ in the 3'UTR of *Ago2* mRNA did not alter Ago2 level (*Figure 2—figure supplement 1F*), indicating that this Trim71-binding site does not regulate Ago2 mRNA translation in cis and is dependent on Trim71 to regulate Ago2 expression.

Second, polysome analysis indicated that *Ago2* mRNA, but not other Trim71 target mRNAs (e.g., *Cdkn1a* mRNA) nor a control mRNA (*Gapdh* mRNA), showed increased ribosome association in the CLIPΔ mESCs compared to WT mESCs (*Figure 2D and E*), indicating translational upregulation.

Third, when ectopically expressed in mESCs, Trim71 did not decrease *Ago2* mRNA level, but reduced Ago2 protein level (*Figure 2F and H*). Moreover, the ectopically expressed Trim71 shifted *Ago2* mRNA from the polysome region to the RNP region on the sucrose density gradient (*Figure 2G*), indicating translation inhibition. This repression is specific to *Ago2* mRNA, as neither Ago1 level (*Figure 2H*) nor the ribosome association of *Gapdh* mRNA (*Figure 2G*) altered when Trim71 was overexpressed.

Fourth, the repression of Ago2 is dependent on Trim71's binding to *Ago2* mRNA, as this repression was lost in CLIPΔ mESCs (*Figure 2I*), where Trim71 does not bind *Ago2* mRNA (*Figure 2—*

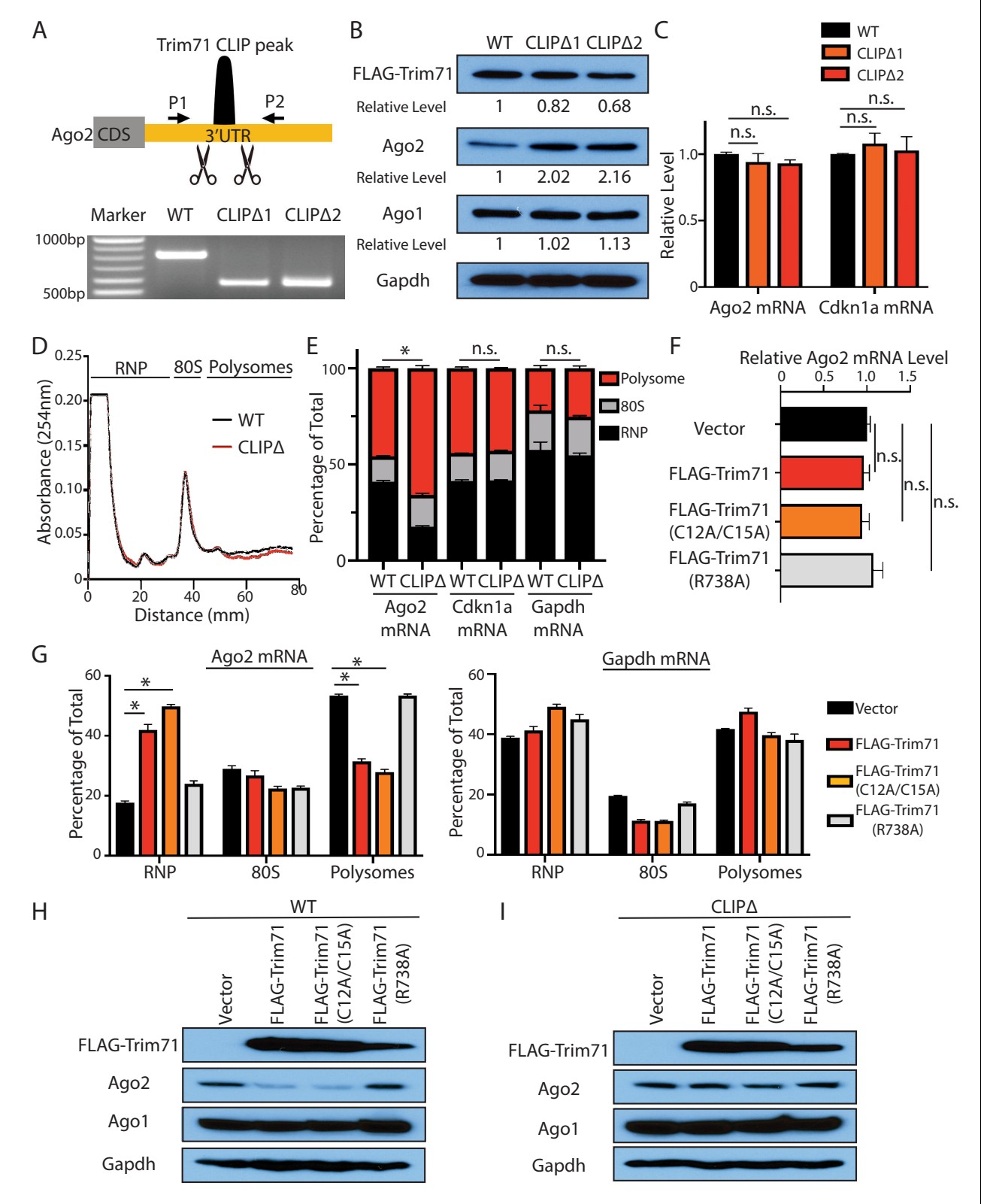

**Figure 2.** Trim71 represses *Ago2* mRNA translation in mouse embryonic stem cells (mESCs). (**A**) Deletion of Trim71 binding region in Ago2 mRNA's 3'UTR. Genotyping PCR was performed using the indicated P1 and P2 primers. CLIPΔ1 and CLIPΔ2 are two independent clones from the genomic editing. (**B**) Western blotting in the WT, CLIPΔ1, and CLIPΔ2 mESCs. (**C**) qRT-PCR quantification of two Trim71 target mRNAs, *Ago2* mRNA, and *Cdkn1a* mRNA, in the WT, CLIPΔ1, and CLIPΔ2 mESCs. 18S rRNA was used for normalization. (**D**) Polysome analysis in WT and CLIPΔ mESCs. (**E**) Inhibiting

*Figure 2 continued on next page*

*Figure 2 continued*

Trim71's binding on *Ago2* mRNA specifically upregulates its translation. The mRNA distribution in the RNP, the 80S, and the polysome fractions (shown in **C**) were quantified by qRT-PCR in the WT and the CLIPΔ mESCs, respectively. (**F**) Overexpression of Trim71 and its mutants does not change *Ago2* mRNA level in the WT mESCs. The expression level of *Ago2* mRNA in the WT mESCs with an empty vector, FLAG-Trim71, FLAG-Trim71(C12A/C15A), and FLAG-Trim71(R738A) was quantified by qRT-PCR. 18S rRNA was used for normalization. (**G**) Quantification of the indicated mRNA distributions in the RNP, 80S, and polysome fractions in the cell lysates from the WT mESCs expressing an empty vector, FLAG-Trim71, FLAG-Trim71(C12A/C15A), or FLAG-Trim71(R738A). (**H**) Western blotting in WT mESCs expressing an empty vector, FLAG-Trim71, a Trim71 ubiquitination mutant (C12A/C15A), and a Trim71 RNA-binding mutant (R738A). (**I**) Western blotting in CLIPΔ mESCs expressing an empty vector, FLAG-Trim71, a Trim71 ubiquitination mutant (C12A/C15A), and a Trim71 RNA-binding mutant (R738A). The qPCR results in (**C**) and (**E–G**) represent the means (± SD) of three independent experiments. *$p<0.05$, and n.s. not significant ($p>0.05$) by the Student's t-test.

The online version of this article includes the following figure supplement(s) for figure 2:

**Figure supplement 1.** Specific disruption of the interaction between Trim71 and *Ago2* mRNA.

*figure supplement 1D*). Moreover, an RNA-binding mutation (R738A) of Trim71 abolished its ability to repress Ago2 mRNA translation (*Figure 2F–H*).

Lastly, the E3 ligase mutations in Trim71 (C12A/C15A) did not abolish the translation repression of *Ago2* mRNA (*Figure 2F–H*), arguing that Trim71 does not regulate Ago2 through protein degradation in mESCs.

Collectively, these results reveal that the Trim71 represses *Ago2* mRNA translation in mESCs.

## Repressing *Ago2* mRNA translation by Trim71 is required for maintaining stemness

To determine the significance of the Trim71:*Ago2*–mRNA interaction to ESC biology, we compared the WT and the CLIPΔ mESCs' capacities in proliferation, self-renewal, and differentiation.

WT and CLIPΔ mESCs had no morphological difference and proliferated at similar rates (*Figure 2—figure supplement 1G*). However, when self-renewal was evaluated using the colony formation assay, CLIPΔ mESCs displayed a defect in maintaining stemness (*Figure 3A*). When subjected to the exit pluripotency assay, which determines the rate ESCs exit the pluripotent state (*Betschinger et al., 2013*), CLIPΔ mESCs had an increased rate of losing pluripotency (*Figure 3B*). These observations indicated that CLIPΔ mESCs have stemness defects and are prone to differentiation.

To measure differentiation kinetics, we harvested mESCs at various time points during embryonic body (EB) formation. Western blotting revealed that CLIPΔ mESCs showed a faster decline in the levels of all three core pluripotency transcription factors, Nanog, Oct4, and Sox2, compared with WT mESCs (*Figure 3C*). When mESCs were subject to spontaneous monolayer differentiation, structural markers for lineage-committed cells from the three germ layers were detected first and at higher levels in cells from CLIPΔ mESCs compared to WT mESCs (*Figure 3D*). These results indicated that the CLIPΔ mESCs undergo differentiation more rapidly.

The stemness and differentiation defects in the CLIPΔ mESCs are dependent on *Ago2*, as they were lost in the *Ago2* KO genetic background (*Figure 3*). These observations indicate that Trim71-mediated repression of *Ago2* mRNA translation, which is lost in the CLIPΔ mESCs, is required for maintaining stemness in mESCs.

## Inhibiting Trim71-mediated repression of *Ago2* mRNA translation results in a specific post-transcriptional increase of let-7 miRNAs

Ago2 is a key component in the miRNA pathway (*Bartel, 2018*; *Carthew and Sontheimer, 2009*). To determine whether the stemness defects in the CLIPΔ mESCs are dependent on the miRNA pathway, we blocked the miRNA pathway by knocking out *Dicer* or *Dgcr8* (*Figure 3—figure supplement 1A*), which are required for processing pre-miRNAs and pri-miRNAs, respectively (*Ha and Kim, 2014*). In either *Dicer* KO or *Dgcr8* KO mESCs, both mature miRNA levels and miRNA activities were significantly reduced (*Figure 3—figure supplement 1B and C*). In either the *Dicer* KO or the *Dgcr8* KO genetic background, inhibiting the Trim71:*Ago2*–mRNA interaction did not alter mESC self-renewal or differentiation, as determined by colony formation assay and EB differentiation, respectively (*Figure 3—figure supplement 1D–F*). These results indicate that the stemness defects in the CLIPΔ mESCs are dependent on the miRNA pathway.

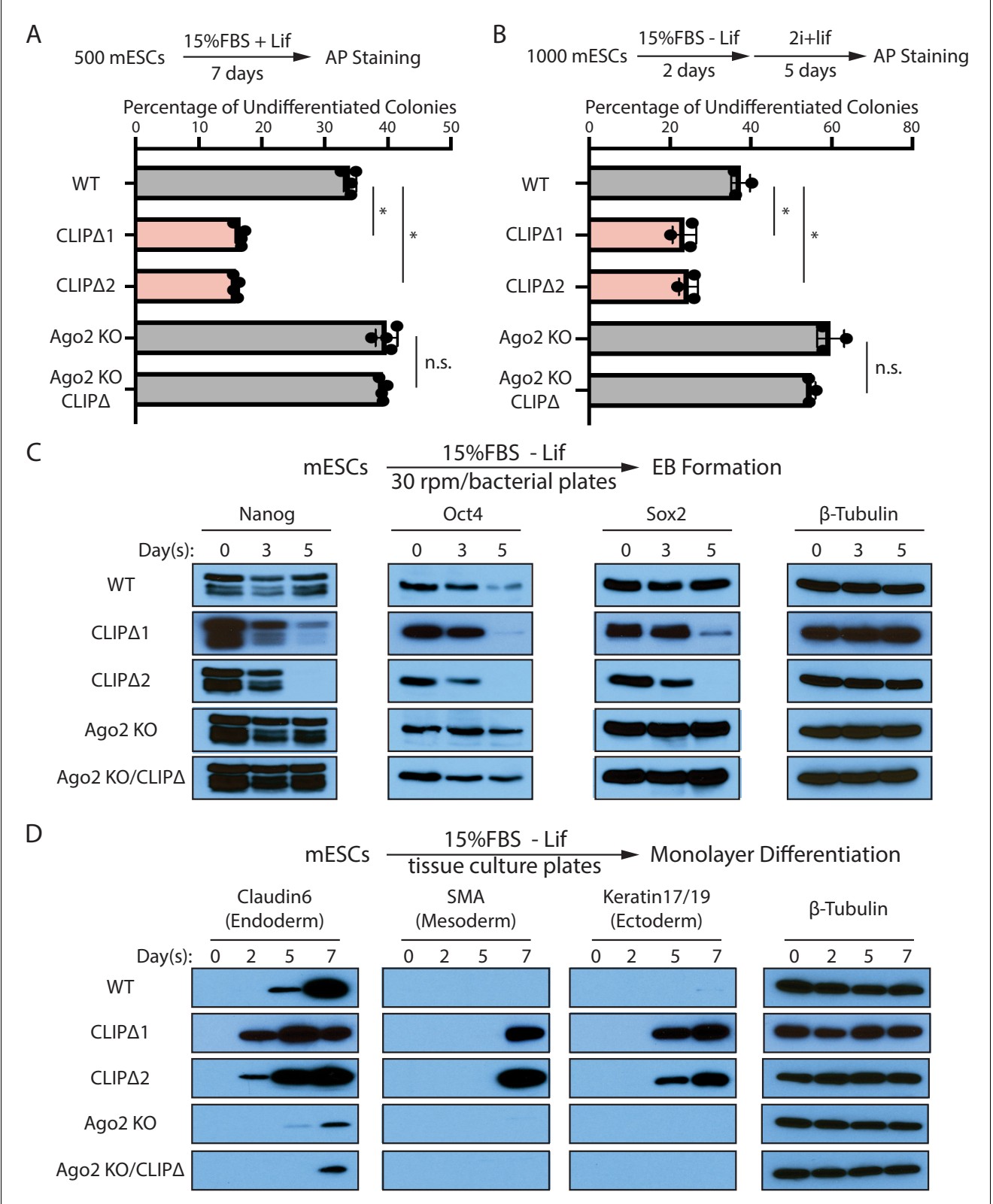

**Figure 3.** Trim71-mediated repression of *Ago2* mRNA translation is required for maintaining pluripotency. (**A**) Colony formation assay for mouse embryonic stem cells (mESCs). The mESCs were cultured in 15%FBS + Lif for 7 days, and the resultant colonies were fixed and stained for AP. (**B**) Exit pluripotency assay for mESCs. The mESCs were induced to exit pluripotency in medium without Lif for 2 days and then switched to 2i+Lif medium for 5 days. The resultant colonies were fixed and stained for AP. In (**A**) and (**B**), the colony morphology and AP intensity were evaluated

*Figure 3 continued on next page*

*Figure 3 continued*

through microscopy. 100–200 colonies were examined each time to determine the percentage of undifferentiated colonies. The results represent the means (± SD) of four independent experiments. *p<0.05, and n.s. not significant (p>0.05) by the Student's t-test. (**C**) Western blotting of pluripotency factors during EB formation. (**D**) Western blotting of markers of lineage-committed cells during mESC monolayer differentiation.

The online version of this article includes the following figure supplement(s) for figure 3:

**Figure supplement 1.** The stemness defects caused by the loss of Trim71-mediated repression of *Ago2* mRNA translation is dependent on the miRNA pathway.

To determine how miRNAs were altered in the CLIPΔ mESCs, we performed small RNA sequencing. We found that WT and CLIPΔ mESCs have similar miRNA expression patterns (*Figure 4A*, *Figure 4—figure supplement 1A and B*). Of the 515 detected miRNAs, only 59 were differentially expressed (*Figure 4A*, *Supplementary file 2*). Interestingly, however, the let-7 miRNAs were the most dramatically increased miRNAs in the CLIPΔ mESCs (*Figure 4A*). We verified this result by qRT-PCR. In the CLIPΔ mESCs, most let-7 miRNAs increased greater than fourfold compared to those in the WT mESCs, while the levels of several non-let-7 miRNAs did not increase (*Figure 4B*). This specific increase of let-7 miRNAs occurs at the post-transcriptional level, as several pri-let-7 miRNAs were not elevated in the CLIPΔ mESCs (*Figure 4C*). Although several pre-let-7 miRNAs were elevated in the CLIPΔ mESCs (*Figure 4C*), the twofold to threefold increase in pre-miRNAs was not at the same magnitude as the increased mature let-7 miRNAs (*Figure 4B and C*), suggesting that let-7 miRNAs are also regulated at the mature miRNA level.

Let-7 miRNAs are conserved pro-differentiation miRNAs that are induced during ESC differentiation (*Büssing et al., 2008*). The following observations, however, indicated that the differentiation program was not activated in the CLIPΔ mESCs. First, all the mESCs for these gene profiling experiments were cultured in 2i+lif medium, a stringent condition for suppressing differentiation and maintaining stemness (*Ying et al., 2008*). Second, except for the let-7 miRNAs, the miRNA expression patterns were highly similar between the WT and the CLIPΔ mESCs, and the expression of mESC-specific miR-290–295 members was not altered (*Figure 4—figure supplement 1C*). Third, the CLIPΔ mESCs expressed similar levels of the pluripotency factors as the WT mESCs (*Figure 4D*), and the markers of the lineage-committed cells were absent at the start of differentiation (*Figure 3D*). Collectively, these results indicated that the increased let-7 miRNAs in the CLIPΔ mESCs were not caused by differentiation.

Consistent with the increased let-7 miRNA levels, the endogenous targets of let-7 miRNAs were repressed in the CLIPΔ mESCs. Western blotting indicated that CLIPΔ mESCs had decreased Trim71 and Lin28a, two conserved targets of the let-7 miRNAs, compared to the WT mESCs; the pluripotency factors (Nanog, Oct4, Sox2), which are not targeted by the let-7 miRNAs (*Melton et al., 2010*), however, were expressed at similar levels (*Figure 4D*). Transcriptomic profiling of the WT and the CLIPΔ mESCs via RNAseq revealed that let-7 target mRNAs, as determined by TargetScan (*Agarwal et al., 2015*), were significantly repressed in the CLIPΔ mESCs compared to non-let-7 miRNAs' targets or mRNAs not targeted by the expressed miRNAs (*Figure 4E*). These results indicated a specific increase of let-7 miRNA activity in the CLIPΔ mESCs.

The increased let-7 miRNA levels and activity are dependent on Ago2, because both the increased let-7 miRNA levels and the repression of let-7 targets were abolished in the CLIPΔ mESCs in the *Ago2* KO genetic background (*Figure 4B and D*).

## Increasing Ago2 levels results in a specific elevation of let-7 miRNAs and stemness defects in mESCs

To determine whether the increased Ago2 leads to the specific increase of let-7 miRNAs and the stemness defects as we observed in the CLIPΔ mESCs, we made stable WT mESC lines, in which Ago2 can be induced by doxycycline (dox) in a dosage-dependent manner (*Figure 5A*). To examine the early effects of increased Ago2 levels on let-7 miRNAs, we performed qRT-PCR on the WT mESCs treated with increasing amounts of dox for 16–20 hr. Most of the let-7 miRNAs showed an Ago2-level-dependent increase, and some of these miRNAs (e.g., let-7f and miR-98) increased approximately 10-fold when Ago2 level was increased approximately fivefold (*Figure 5A and B*, *Figure 5—figure supplement 1A*). The levels of a group of non-let-7 miRNAs, however, were not

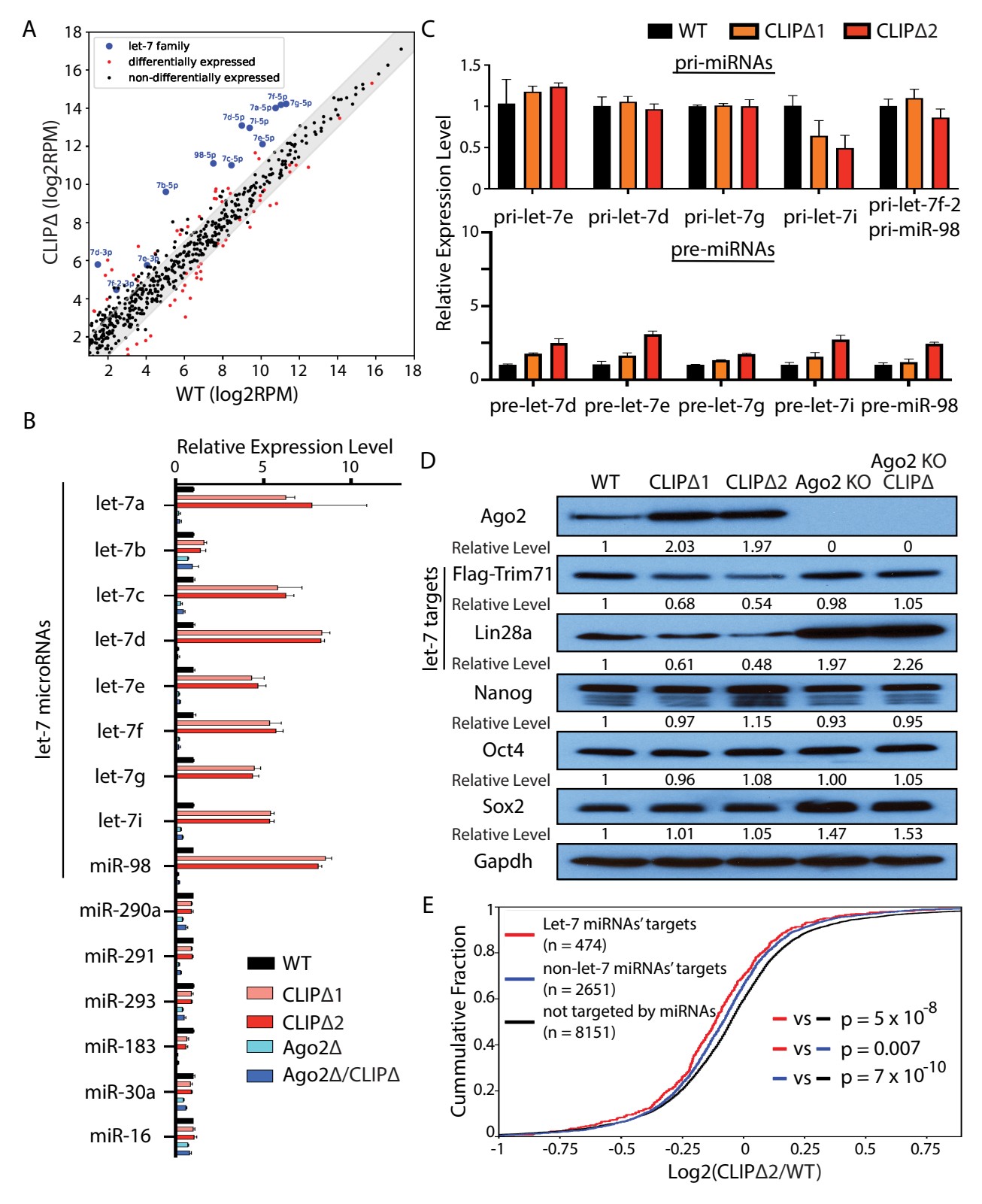

**Figure 4.** Loss of Trim71-mediated repression of *Ago2* mRNA translation results in significant post-transcriptional increase of let-7 miRNAs. (**A**) Comparison of global miRNA expression in WT and CLIPΔ mouse embryonic stem cells (mESCs). The results are the average of four independent small RNA-seqs in the WT and the CLIPΔ mESCs. Blue dots: let-7 miRNAs; red dot: differentially expressed miRNAs; black dots: non-differentially expressed miRNAs. (**B**) qRT-PCR on let-7 miRNAs and non-let-7 miRNAs. For each miRNA, the expression level in WT cells was set as 1 for relative comparison. U6

*Figure 4 continued on next page*

*Figure 4 continued*

RNA was used for normalization. (**C**) qRT-PCR on the let-7 pri-miRNAs and pre-miRNAs. For pri-miRNAs and pre-miRNAs, the expression level in the WT cells was set as 1 for relative comparison. 18S rRNA and U6 RNA were used for pri-miRNA and pre-miRNA normalization, respectively. The results in (**B**) and (**C**) are from three independent replicates. (**D**) Western blotting of Ago2, conserved let-7 targets, and non-let-7 targets. Gapdh was used for normalization in calculating the relative expression levels. (**E**) Cumulative distributions of expression level changes of let-7 targets, miRNA targets without let-7 binding sites, and mRNAs not targeted by miRNAs in WT and CLIPΔ mESCs.

The online version of this article includes the following figure supplement(s) for figure 4:

**Figure supplement 1.** The loss of Trim71-mediated repression of *Ago2* mRNA translation does not alter global miRNA in mouse embryonic stem cells (mESCs).

elevated at these increasing amounts of Ago2 (*Figure 5B* and *Figure 5—figure supplement 1A*), indicating that increasing Ago2 level in mESCs results in a specific increase of let-7 miRNAs.

In mammals, miRNAs can associate with all the four Ago proteins. To examine whether the increase of let-7 miRNAs is specific to Ago2, we increased the level of another Ago protein, Ago1, which is expressed in mESCs (*Figure 5—figure supplement 1B and C*). Similar to the results from Ago2, increasing Ago1 level also resulted in a specific post-transcriptional increase of let-7 miRNAs in mESCs (*Figure 5—figure supplement 1D and E*).

Consistent with increased let-7 miRNAs, the two conserved let-7 targets, Trim71 and Lin28a, displayed decreasing levels in the mESCs with increasing amounts of Ago2, while the levels of non-let-7 targets, such as Nanog, Oct4, and Sox2, were not altered in these mESCs (*Figure 5A*). This Ago2-mediated increase of let-7 miRNAs occurred at the post-transcriptional level because the let-7 pri-miRNAs were not elevated in the mESCs with increasing amounts of Ago2 (*Figure 5C*).

To evaluate whether increasing Ago2 results in stemness defects, we examined the ability to maintain stemness and the rate of exit pluripotency by the colony formation assay and the exit pluripotency assay, respectively, in the mESCs with increasing amounts of Ago2. When the Ago2 level was elevated, the mESCs had decreased ability in maintaining stemness and increased rates in exit pluripotency (*Figure 5D and E*). Consistent with these, increased Ago2 resulted in a faster decline in the levels of the pluripotency factors (e.g., Nanog and Oct4) during EB formation (*Figure 5F*).

Collectively, these results argued that the specific increase of let-7 miRNAs and the stemness defects in the CLIPΔ mESCs are caused by the increased Ago2.

## The increased let-7 miRNAs are bound and stabilized by Ago2 in mESCs

Ago2 binds all miRNAs. Why does the elevation of Ago2 result in a specific increase of let-7 miRNAs in mESCs? A unique aspect of the pro-differentiation let-7 miRNAs in mESCs is that although genes encoding let-7 miRNAs are actively transcribed (*Suh et al., 2004*; *Thomson et al., 2006*), the let-7 miRNA levels are low, indicating post-transcriptional regulations of let-7 miRNAs. Indeed, the processing of let-7 pre-miRNAs are repressed by Lin28a in mESCs (*Hagan et al., 2009*; *Heo et al., 2008*). Since forming miRNPs (miRNA–protein complex) with Ago2 stabilizes mature miRNAs (*Diederichs and Haber, 2007*; *Winter and Diederichs, 2011*), we speculated that increased Ago2 in mESCs stabilizes the over-produced let-7 miRNAs that are degraded when Ago2 level is limiting. To test this, we performed two experiments.

We first determined that the increased let-7 miRNAs are bound by Ago2 in the Ago2 elevated mESCs. We generated mESCs with a FLAG-tag at the N-terminus of the endogenous Ago2, which enabled specific isolation of both the endogenous Ago2 and the dox-induced FLAG-Ago2 via the anti-FLAG antibody (*Figure 5—figure supplement 2A*). RNA immunoprecipitation and qRT-PCR revealed that among the Ago2-bound miRNAs, let-7 miRNAs are specifically increased when Ago2 is elevated (*Figure 5—figure supplement 2B and C*). Then, we measured RNA stability through actinomycin-D-mediated transcriptional shut-off (*Figure 5—figure supplement 2D*). We found that increased Ago2 did not alter the decay of let-7 pri-miRNAs, but specifically stabilized the let-7 miRNAs (*Figure 5—figure supplement 2E and F*).

Collectively, these observations not only indicate that the increased Ago2 directly binds and protects let-7 miRNAs from degradation in mESCs, but also argue that besides the previously characterized Lin28a-mediated inhibition of pre-miRNA processing, let-7 miRNAs are also repressed at the

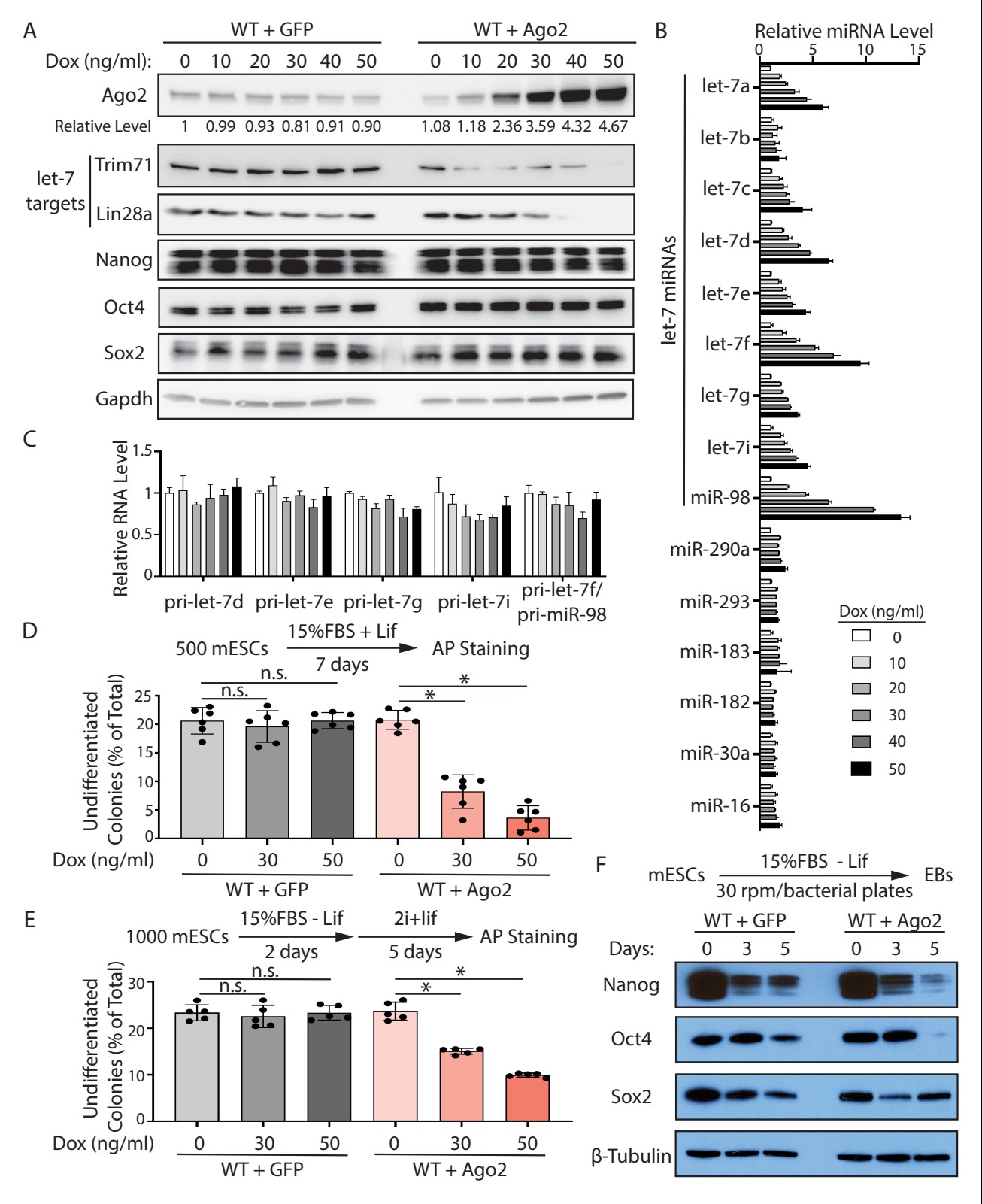

**Figure 5.** Increased Ago2 leads to significant increase of let-7 miRNAs and accelerated differentiation in mouse embryonic stem cells (mESCs). (**A**) Western blotting in mESCs with dox-inducible expression of Ago2. Gapdh was used for normalization in calculating the relative expression of Ago2. (**B**) Relative levels of miRNAs in mESCs with dox-inducible expression of Ago2. U6 RNA was used for normalization. (**C**) Relative levels of pri-miRNAs in mESCs with dox-inducible expression of Ago2. 18S rRNA was used for normalization. In (**B**) and (**C**), the miRNA and pri-miRNA expression levels in

*Figure 5 continued on next page*

*Figure 5 continued*

mESCs without dox treatment were set as 1 for determining relative levels. The results are from four biological replicates. (D) Colony formation assay for mESCs with dox-inducible expression of either GFP or Ago2. (E) Exit pluripotency assay for mESCs with dox-inducible expression of either GFP or Ago2. The results in (D) and (E) represent the means (± SD) of six independent experiments. *p<0.05, and n.s. not significant (p>0.05) by the Student's t-test. (F) Western blot analysis on pluripotency factors during EB formation from the GFP-expressing mESCs and Ago2-expressing mESCs (treated with 50 ng/ml dox).

The online version of this article includes the following figure supplement(s) for figure 5:

**Figure supplement 1.** Increased Ago proteins in mouse embryonic stem cells (mESCs) result in a specific increase of let-7 miRNAs.

**Figure supplement 2.** The increased let-7 miRNAs are bound and stabilized by the elevated Ago2 in mouse embryonic stem cells (mESCs).

mature miRNA level in mESCs by the limiting Ago2 level (e.g., caused by the Trim71-mediated repression of *Ago2* mRNA translation).

## The stemness defects in the CLIPΔ mESCs are dependent on the let-7 miRNAs

Let-7 miRNAs have a conserved function in promoting cell differentiation (*Büssing et al., 2008*; *Lee et al., 2016*; *Roush and Slack, 2008*). To determine whether the stemness defects caused by the increased Ago2 in the CLIPΔ mESCs is dependent on the let-7 miRNAs, we performed the following experiments.

First, we repressed let-7 miRNAs through generating stable mESC lines in which the expression of exogenous Lin28a, Lin28b, or GFP can be induced by dox. Lin28a and Lin28b specifically repress the maturation of let-7 miRNAs at the pre-miRNA and pri-miRNA levels, respectively (*Hagan et al., 2009*; *Heo et al., 2008*; *Piskounova et al., 2011*). *Lin28a*, but not *Lin28b*, is highly expressed in mESCs (*Figure 6—figure supplement 1A*). When either Lin28a or Lin28b was induced (*Figure 6A*), the let-7 miRNAs in the CLIPΔ mESCs were reduced to levels similar to those in the WT mESCs with no significant alterations in a group of non-let-7 miRNAs (*Figure 6B*). Consistent with this, the activities of let-7 miRNAs were also specifically repressed. The levels of the conserved let-7 targets, Trim71 and Lin28a, increased in the CLIPΔ mESCs when the exogenous Lin28a or Lin28b was expressed, but the levels of non-let-7 targets (e.g., Nanog, Oct4, and Sox2) were not altered (*Figure 6A*). The ectopically expressed Lin28a or Lin28b alleviated the decreased ability to maintain stemness and inhibited the increased rate of exit pluripotency in the CLIPΔ mESCs, as determined by the colony formation assay and the exit pluripotency assay, respectively (*Figure 6C and D*). Moreover, western blotting revealed that the ectopically expressed Lin28a or Lin28b also inhibited the rapid decrease of pluripotency factors during EB formation in the CLIPΔ mESCs (*Figure 6E*). These results argued that the stemness defects in the CLIPΔ mESCs are dependent on the increased let-7 miRNAs.

One caveat of the Lin28 ectopic expression is the potential pleiotropic effects (reviewed in *Tsialikas and Romer-Seibert, 2015*). To address this and to specifically determine whether let-7 miRNAs are responsible for the stemness defects, in a parallel experiment, we used locked nucleic acid antisense oligonucleotides (LNA) targeting the conserved seed sequence of let-7 miRNAs to attenuate their activities (*Figure 6—figure supplement 1B*). In the presence of the anti-let-7 LNA, the CLIPΔ mESCs had an increase (threefold) in the ability of maintaining stemness as determined by the colony formation assay (*Figure 6F*), indicating that the decreased stemness in the CLIPΔ mESCs is dependent on let-7 miRNAs.

Altogether, the results from the ectopic expression of Lin28a/b and LNA-mediated inhibition of let-7 miRNAs indicated that the stemness defects caused by the loss of Trim71-mediated repression of *Ago2* mRNA translation are dependent on the increased let-7 miRNAs.

## Trim71 represses mRNA translation at post-initiation step(s) in mESCs

Trim71 can repress mRNA translation (*Aeschimann et al., 2017*; *Loedige et al., 2013*). Recent observations, however, argued that Trim71 inhibits gene expression through mRNA degradation in mESCs (*Welte et al., 2019*). To determine whether or not the Trim71-mediated translation repression is mRNA-specific in mESCs (e.g., *Ago2* mRNA in *Figure 2*), we performed the tethering assay. When tethered to a FLuc mRNA via the specific interaction between the bacteriophage λN

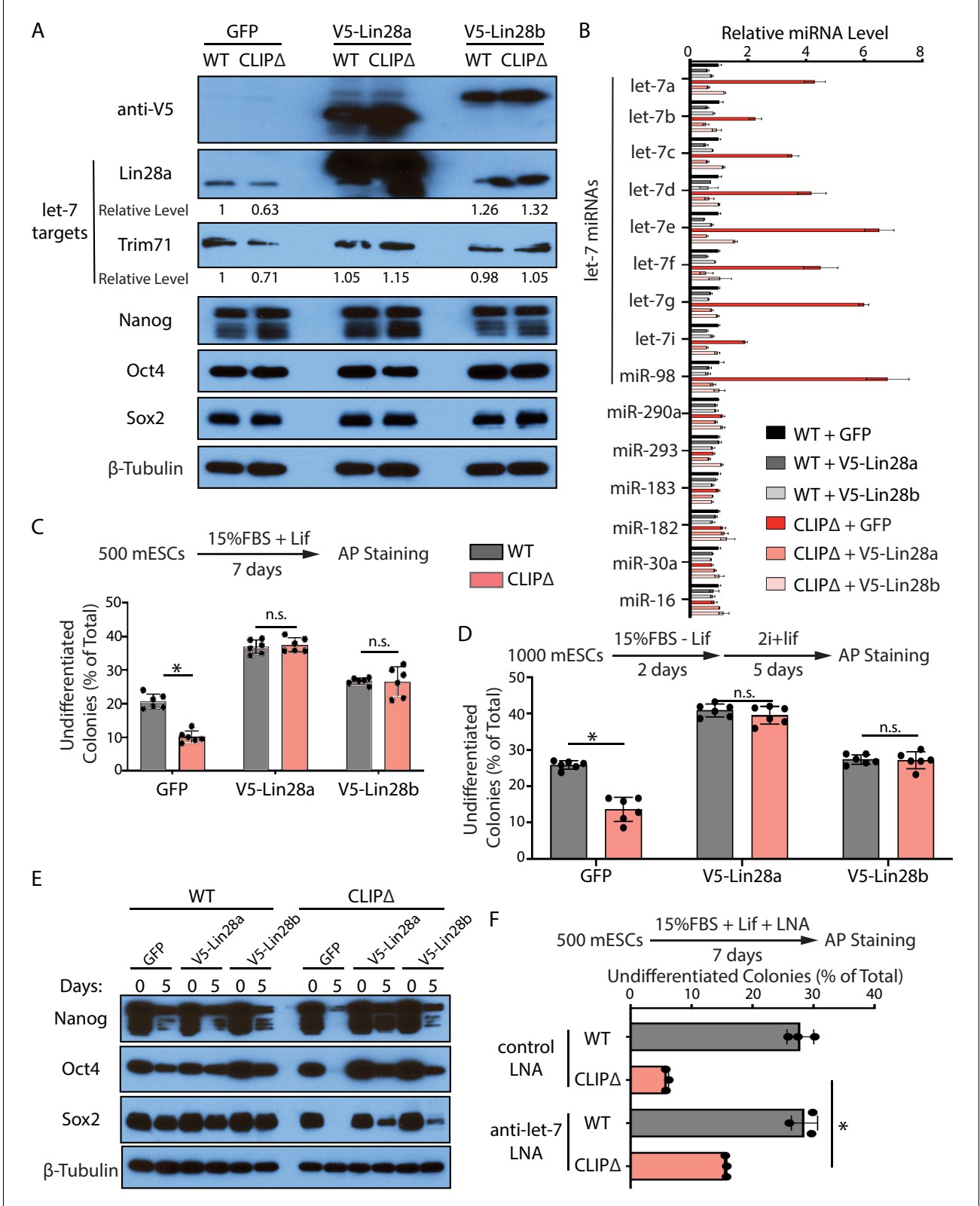

**Figure 6.** Inhibiting let-7 miRNAs blocks the stemness defects caused by the loss of Trim71-mediated repression of *Ago2* mRNA translation. (**A**) Western blotting in WT and CLIPΔ mouse embryonic stem cells (mESCs) expressing GFP, V5-Lin28a, or Lin28b. Beta-tubulin was used for normalization in determining the relative expression level of let-7 targets Lin28a and Trim71. (**B**) Relative levels of miRNAs. U6 RNA was used for normalization. The results represent the means (± SD) of four biological replicates. (**C**) Colony formation assay for WT and CLIPΔ mESCs expressing GFP, V5-Lin28a, or

*Figure 6 continued on next page*

*Figure 6 continued*

Lin28b. (D) Exit pluripotency assay for WT and CLIPΔ mESCs expressing GFP, V5-Lin28a, or Lin28b. The results in (C) and (D) represent the means (± SD) of six independent experiments. (E) Western blotting of pluripotency factors during EB formation at Day 0 and Day 5 of WT and CLIPΔ mESCs expressing GFP, V5-Lin28a, or Lin28b. (F) Colony formation assay for WT and CLIPΔ mESCs cultured in the presence of 500 nM anti-let-7 LNA or a control LNA. The results represent three independent experiments. *p<0.05, and n.s. not significant (p>0.05) by the Student's t-test.

The online version of this article includes the following figure supplement(s) for figure 6:

**Figure supplement 1.** Inhibition of let-7 miRNAs in mouse embryonic stem cells (mESCs).

polypeptide and the BoxB RNA motif, Trim71 reduced the FLuc activity but not the FLuc mRNA level (*Figure 7A and B*), indicating translation repression. This repression is specific, as Trim71 does not repress the control mRNA without the BoxB sites (*Figure 7B*). Thus, repressing mRNA translation can be a general mechanism for Trim71 in mESCs.

To determine how Trim71 represses translation in mESCs, we used bicistronic reporters containing either the HCV-IRES (internal ribosome entry site), which requires all the initiation factors except eIF4G and eIF4E, or the CrPV-IRES, which only requires the 40S ribosomal subunit for initiating translation (*Fraser and Doudna, 2007*; *Figure 7C*). In these reporters, FLuc was produced by the canonical translation, and the RLuc was generated by the IRES-mediated translation. Tethering Trim71 to either of these two reporter mRNAs led to a decrease of both FLuc and RLuc activities, while no changes in mRNA levels (*Figure 7D and E*). This result indicated that Trim71 either interferes with 40S ribosomal subunit recruitment or inhibits an event at or after the 60S subunit joining step during mRNA translation. Moreover, we found that Trim71-mediated translation repression does not require 3′ end poly(A) tail. When tethered to a FLuc mRNA that is devoid of both poly(A) tail and the poly(A) tail binding protein, Pabpc1 (*Figure 7F*; *Zhang et al., 2017*), Trim71 also specifically represses the reporter mRNA translation (*Figure 7G and H*). Since poly(A) tail and Pabpc1 can promote mRNA translation at multiple steps, including 40S ribosomal subunit recruitment and the 60S ribosomal subunit joining step during the initiation process (*Kahvejian et al., 2005*; *Mangus et al., 2003*), these observations, combined with the result from the IRES reporters, argue that Trim71 regulates mRNA translation at a post-initiation step(s) in mESCs.

## Discussion

Our data reveal that Trim71 maintains pluripotency in stem cells by specifically inhibiting the conserved let-7 miRNAs through repressing *Ago2* mRNA translation. These results not only provide direct support for the conserved cytoplasmic bi-stable switch model (*Ecsedi and Grosshans, 2013*) in stem-cell fate decisions, but also revealed that a new layer of regulation on the conserved pro-differentiation let-7 miRNAs: repressing the mature miRNA by Ago2 availability. This regulation is critical for pluripotency in stem cells. Our findings raise several interesting aspects in stem cell biology and RNA biology.

### Ago2 and let-7 miRNAs

Previous studies indicate that overexpressing Ago2 in certain cells (e.g., 293T and NIH3T3) elevated global miRNA levels by stabilizing mature miRNAs (*Diederichs and Haber, 2007*; *Winter and Diederichs, 2011*). Our results, however, indicate that elevated Ago2 specifically increased the let-7 miRNAs in mESCs at the post-transcriptional level. Although different cell types may contribute to these different results, an important variable is the Ago2 level. Transfection-based assays tend to result in high expression of exogenous genes. In our CLIPΔ mESCs or the WT mESCs with the dox-induced exogenous Ago2, however, the Ago2 level increased approximately twofold or maximally approximately fivefold, respectively, compared to that in the WT mESCs. This modest increase is biologically relevant because the Trim71-mediated repression of *Ago2* mRNA translation only has an approximately twofold effect on the Ago2 level in mESCs (*Figure 2*). When this approximately twofold repression on Ago2 was specifically disrupted, the mESCs displayed let-7-miRNA-dependent defects in maintaining pluripotency (Figures 3 and 6). Thus, although previous studies (*Diederichs and Haber, 2007*; *Winter and Diederichs, 2011*) and ours all indicate that Ago2 is the limiting factor in forming the effector miRNPs, modulating Ago2 levels under

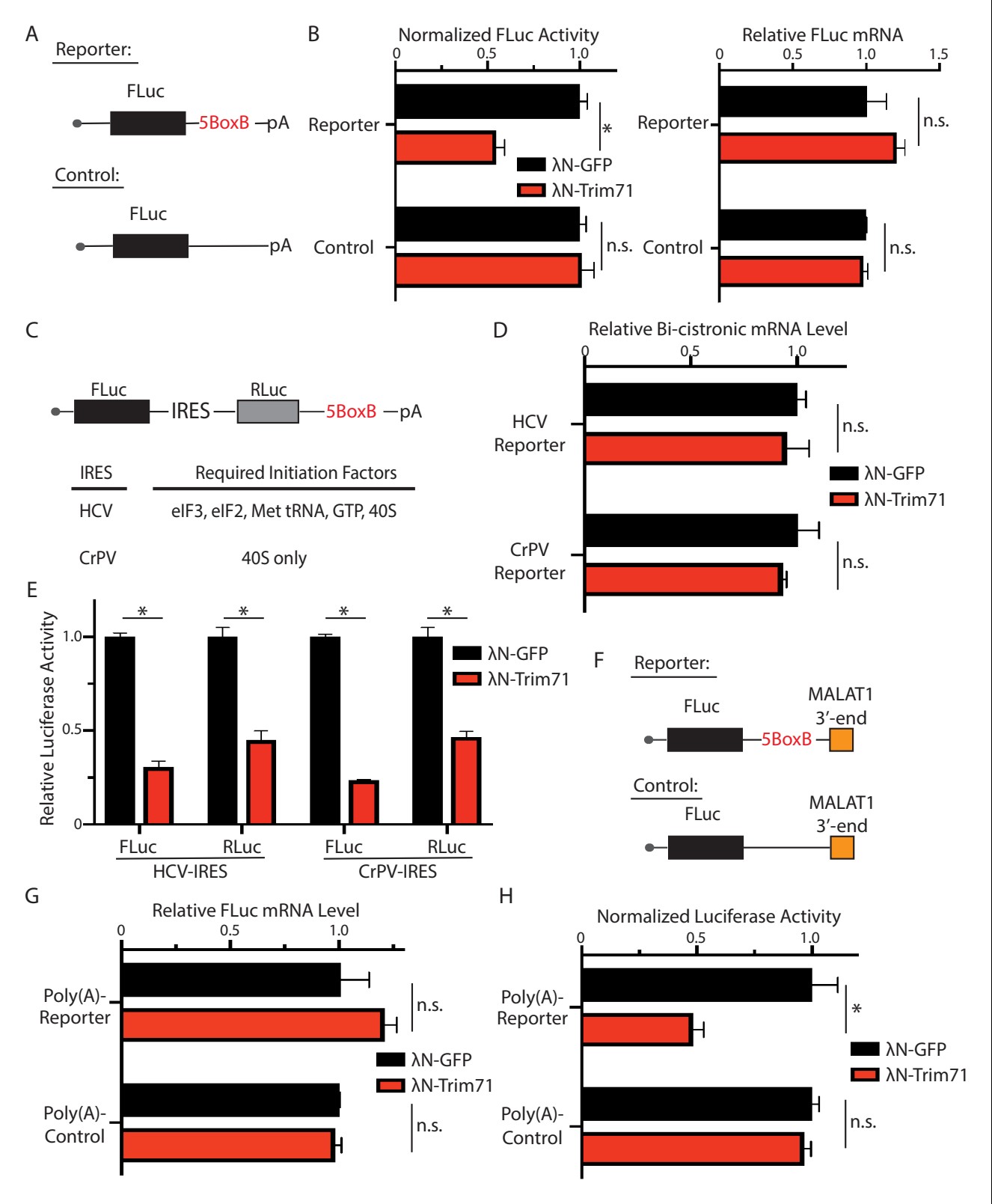

**Figure 7.** Trim71 represses mRNA translation at post-initiation step(s) in mouse embryonic stem cells (mESCs). (**A**) FLuc reporters for the tethering assay. (**B**) The FLuc activity and mRNA level determined in the tethering assay. (**C**) The IRES-containing bicistronic reporters. (**D**) mRNA levels from the IRES-containing reporters. (**E**) Luciferase activities from the IRES-containing reporters. (**F**) The poly(A) minus FLuc reporters. (**G**) mRNA levels from the

*Figure 7 continued on next page*

*Figure 7 continued*

poly(A) minus reporters. (**H**) FLuc activities from the poly(A) minus reporters. The results represent the means (± SD) of three independent experiments. *p<0.05, and ns. not significant (p>0.05) by the Student's t-test.

biologically relevant settings is likely to result in changes of specific miRNAs, such as the let-7 miR-NAs. Ago2 and let-7 miRNAs levels are frequently dysregulated in cancers (*Ye et al., 2015*). We speculate that modulating Ago2 levels may also regulate tumorigenesis by altering the activities of specific miRNAs (e.g., the tumor-suppressive let-7 miRNAs).

## Two inter-connected layers of regulations on let-7 miRNAs in stem cells

Let-7 is a conserved pro-differentiation miRNA that is abundantly expressed in differentiated cells (*Roush and Slack, 2008*). In stem cells, although the genes encoding let-7 miRNAs are actively transcribed, the levels of mature let-7 miRNAs are low (*Suh et al., 2004*; *Thomson et al., 2006*), indicating post-transcriptional inhibition of let-7 miRNAs. Previous studies characterized that the conserved RBPs Lin28a and Lin28b inhibit let-7 miRNAs' maturation at the pre- and pri-miRNA levels (*Tsialikas and Romer-Seibert, 2015*). Here, we revealed an additional layer of regulation of let-7 miRNAs at the mature miRNA level by Ago2 availability.

Interestingly, these two layers of regulation on let-7 miRNAs are intertwined through Lin28a, a conserved let-7 target that promotes let-7 pre-miRNA degradation, in mESCs. When the let-7 miR-NAs were elevated by the increased Ago2, in either the CLIPΔ mESCs or the WT with induced Ago2, there was a corresponding decrease of the endogenous Lin28a (*Figure 4D* and *Figure 5A*). Moreover, the decreased Lin28a resulted in a compromise in the degradation of let-7 pre-miRNAs, as indicated by twofold to threefold increase of let-7 pre-miRNAs in the CLIPΔ mESCs (*Figure 4C*). This compromise explains the significant increase of mature let-7 miRNAs upon a modest increase of Ago2 (*Figure 4* and *Figure 5*): the initial increase of the let-7 miRNAs caused by the elevated Ago2 decreases Lin28a and alleviates Lin28a-mediated inhibition on the maturation of let-7 miRNAs, resulting in more let-7 pre-miRNAs become mature let-7 miRNAs. This positive regulatory loop amplifies let-7 miRNAs and makes the pro-differentiation let-7 miRNAs sensitive to Ago2 levels in stem cells.

The two inter-connected layers of inhibition on the potent pro-differentiation let-7 miRNAs are important to maintaining stemness. During self-renewal of mESCs, although genes encoding let-7 miRNAs are transcribed, the production of these potent pro-differentiation miRNAs is inhibited at both the pre-miRNA level by Lin28a and the mature miRNA level by Ago2. These two layers of regulation may work together to ensure low levels of let-7 miRNPs in stem cells. Considering that let-7 miRNAs are repressed in cancers (*Büssing et al., 2008*), and Lin28a or Lin28b are highly expressed in many cancers (*Piskounova et al., 2011*), we speculate that the Ago2-mediated regulation of let-7 miRNPs may also be employed by cancer cells for their dysregulated proliferation.

## Trim71 and Ago2

The Trim71-mediated downregulation of Ago2 occurs through repressing *Ago2* mRNA translation in mESCs (*Figure 2*), but not the proposed E3-ligase-mediated protein degradation (*Rybak et al., 2009*). This result is consistent with previous observations that the potential E3 ligase activity is not required for Trim71-mediated gene regulation in vivo (*Chen et al., 2012*; *Welte et al., 2019*). In human cells, TRIM71 level negatively correlates with AGO2 level (*Chen et al., 2013*). Thus, we speculate that the Trim71-mediated repression of *Ago2* mRNA translation is conserved between mouse and human. Although the Trim71-binding region in the 3'UTR of mouse *Ago2* mRNA is not conserved in sequence, Trim71 recognizes its RNA targets via structural features but not sequence motifs (*Figure 1G*; *Kumari et al., 2018*; *Welte et al., 2019*). Both human and mouse *Ago2* have long 3'UTRs (5.2 kb and 11.8 kb, respectively) with multiple in silico predicted hairpins that Trim71 can bind. However, our CLIP-seq data indicated that Trim71 only binds one such predicted hairpin in mESCs (*Figure 1I*), suggesting that either not all the predicted hairpins form in vivo or Trim71 uses additional features for target recognition. Trim71 can both repress mRNA translation and induce mRNA degradation (*Aeschimann et al., 2017*; *Loedige et al., 2013*; *Welte et al., 2019*;

*Worringer et al., 2014*). Thus, it will be interesting to determine what features determine whether Trim71 represses translation or destabilizes mRNA.

## A new approach for functional characterization of RBPs

Trim71 is essential for animal development (*Cuevas et al., 2015*; *Ecsedi and Grosshans, 2013*). Previous studies, however, reported no stemness defects in Trim71 knockout mESCs (*Mitschka et al., 2015*; *Welte et al., 2019*). Moreover, Ago2 level was not altered in Trim71 knockdown or knockout mESCs (*Chang et al., 2012*; *Welte et al., 2019*), which we also observed in our mESCs (*Figure 2— figure supplement 1A*). In contrast to these results, our study indicates that Trim71 plays a critical role in regulating pluripotency in mESCs through repressing *Ago2* mRNA translation. How to reconcile these results?

The main difference is the number of disrupted Trim71:mRNA interactions. In the CLIPΔ mESCs, we specifically inhibited one: the Trim71:*Ago2*–mRNA interaction; while in Trim71 knockout/knockdown mESCs, hundreds of Trim71:mRNA interactions and potential Trim71-mediated protein–protein interactions are lost/attenuated. The phenotype of the Trim71 knockout mESCs is the functional additions of all the disrupted interactions. If Trim71's target mRNAs contain both positive and negative regulators of pluripotency, knocking out Trim71 may result in no phenotypical changes. Interestingly, our Trim71 CLIP-seq indicated that besides *Ago2* mRNA, Trim71 also binds mRNAs encoding pluripotency factors, such as Nanog (*Supplementary file 1*). Thus, it is possible that in the Trim71 knockout mESCs, the defects caused by the increased Ago2 may be compensated by an increased Nanog from the lost Trim71:*Nanog*–mRNA interaction. Alternatively, Ago2 protein level does not change in the Trim71 knockout/knockdown mESCs due to combinatory results of the disrupted Trim71:*Ago2* mRNA interaction and secondary effects from other lost Trim71:mRNA interactions, leading to no stemness defects in the knockout mESCs. Thus, an important caveat in interpreting results from the knockout studies on RBPs is that no phenotypical changes does not necessarily mean that the target RBP is not functionally significant. Then, how to effectively characterize biological functions of RBPs?

Thanks to the wide applications of CLIP-based methods, many RBPs' target RNAs and the binding regions in these RNAs are being well characterized. In addition to the loss-of-function methods, we believe specific inhibition of candidate RBP:mRNA interaction(s) via mutating/deleting binding sites on the target mRNA(s) will reveal more exciting roles of RBPs and significant RBP:mRNA interactions under many biological processes.

# Materials and methods

## Key resources table

| Reagent type (species) or resource | Designation | Source or reference | Identifiers | Additional information |
|---|---|---|---|---|
| Antibody | Mouse monoclonal anti-FLAG M2 | Sigma-Aldrich | Cat# F1804 | WB (1:5000) IP |
| Antibody | Normal mouse IgG | Santa Cruz Biotechnology | Cat# sc-2025 | IP |
| Antibody | Mouse monoclonal anti-GAPDH (6C5) | Santa Cruz Biotechnology | Cat# sc-32233 | WB (1:5000) |
| Antibody | Rabbit monoclonal anti-beta-Tubulin | Selleckchem | Cat# A5032 | WB (1:5000) |
| Antibody | Rabbit monoclonal anti-Ago1 (D84G10) | Cell Signaling Technology | Cat# 5053 | WB (1:1000) |
| Antibody | Rabbit monoclonal anti-Ago2 (C34C6) | Cell Signaling Technology | Cat# 2897 | WB (1:1000) |
| Antibody | Mouse monoclonal anti-Oct-4 | BD Transduction Laboratories | Cat# 611202 | WB (1:5000) |
| Antibody | Rabbit monoclonal anti-Nanog (D2A3) | Cell Signaling Technology | Cat# 8822 | WB (1:3000) |

*Continued on next page*

*Continued*

| Reagent type (species) or resource | Designation | Source or reference | Identifiers | Additional information |
|---|---|---|---|---|
| Antibody | Rabbit monoclonal anti-Sox2 (D9B8N) | Cell Signaling Technology | Cat# 23064 | WB (1:3000) |
| Antibody | Rabbit monoclonal anti-Keratin 17/19 (D32D9) | Cell Signaling Technology | Cat# 3984 | WB (1:1000) |
| Antibody | Rabbit monoclonal anti-a-SMA (D4K9N) | Cell Signaling Technology | Cat# 19245 | WB (1:1000) |
| Antibody | Mouse monoclonal anti-Claudin-6 (A-4) | Santa Cruz Biotechnology | Cat# sc-393671 | WB (1:1000) |
| Antibody | Rabbit polyclonal anti-Dicer | Sigma-Aldrich | Cat# SAB4200087 | WB (1:3000) |
| Antibody | Rabbit monoclonal anti-DGCR8 | Abcam | Cat# ab191875 | WB (1:3000) |
| Antibody | Rabbit polyclonal anti-V5 Tag | Bethyl | Cat# A190-120A | WB (1:5000) |
| Antibody | Rabbit monoclonal anti-Lin28A (D1A1A) | Cell Signaling Technology | Cat# 8641 | WB (1:5000) |
| Antibody | Sheep polyclonal anti-Trim71 | R and D Systems | Cat# AF5104 | WB (1:1000) |
| Antibody | Goat Anti-Rabbit IgG (H L)-HRP Conjugate | Bio-Rad | Cat# 170–6515 | WB (1:5000) |
| Antibody | Goat Anti-Mouse IgG (H L)-HRP Conjugate | Bio-Rad | Cat# 170–6516 | WB (1:5000) |
| Antibody | Donkey anti-Sheep IgG-HRP Conjugate | R and D Systems | Cat# HAF016 | WB (1:2000) |
| Chemical compound, drug | DMEM/F-12 | Gibco | Cat# 12500096 | |
| Chemical compound, drug | FBS | Millipore | Cat# ES-009-B | |
| Chemical compound, drug | mLIF | Millipore | Cat# ESG1107 | |
| Chemical compound, drug | PD0325901 | APExBio | Cat# A3013 | |
| Chemical compound, drug | CHIR99021 | APExBio | Cat# A3011 | |
| Chemical compound, drug | N2 | Millipore | Cat# SCM012 | |
| Chemical compound, drug | B27 | Thermo Fisher Scientific | Cat# 17504044 | |
| Chemical compound, drug | MEM NEAA | Gibco | Cat# 11140–50 | |
| Chemical compound, drug | Penicillin–Streptomycin | Gibco | Cat# 11548876 | |
| Chemical compound, drug | L-glutamine | Sigma-Aldrich | Cat# G7513 | |
| Chemical compound, drug | β-mercaptoethanol | Sigma-Aldrich | Cat# M3148 | |
| Chemical compound, drug | Accutase | Millipore | Cat# SF006 | |
| Chemical compound, drug | Fugene6 | Promega | Cat# E2691 | |
| Chemical compound, drug | Puromycin | Sigma-Aldrich | Cat# P9620 | |

*Continued on next page*

Continued

| Reagent type (species) or resource | Designation | Source or reference | Identifiers | Additional information |
|---|---|---|---|---|
| Chemical compound, drug | Doxycycline | Sigma-Aldrich | Cat# D9891 | |
| Chemical compound, drug | Protease inhibitors | Bimake | Cat# B14001 | |
| Chemical compound, drug | Gelatin | Sigma-Aldrich | Cat# G1890 | |
| Chemical compound, drug | One Step-RNA Reagent | Bio Basic | Cat# BS410A | |
| Chemical compound, drug | DNase 1 | NEB | Cat# M0303L | |
| Chemical compound, drug | RNase1 | Ambion | Cat# AM2295 | |
| Chemical compound, drug | SUPERaseIn RNase Inhibitor | Ambion | Cat# AM2696 | |
| Chemical compound, drug | SuperScript II Reverse Transcriptase | Invitrogen | Cat# 18064014 | |
| Chemical compound, drug | SsoAdvanced Universal SYBR Green Supermix | Bio-Rad | Cat# 1725270 | |
| Chemical compound, drug | Q5 High-Fidelity DNA Polymerase | NEB | Cat# M0491L | |
| Chemical compound, drug | Let-7 LNA | Qiagen | Cat# YFI0450006 | |
| Chemical compound, drug | Control LNA | Qiagen | Cat# 339137 | |
| Chemical compound, drug | Actinomycin D | Thermo Fisher Scientific | Cat# 11805017 | |
| Commercial assay or kit | Alkaline Phosphatase Assay Kit | System Biosciences | Cat# AP100R-1 | |
| Commercial assay or kit | Gibson Assembly Master Mix | NEB | Cat# E2611L | |
| Commercial assay or kit | Dual-Luciferase Reporter Assay System | Promega | Cat# E1960 | |
| Commercial assay or kit | CellTiter 96 AQueous One Solution Cell Proliferation Assay (MTS) | Promega | Cat# G3582 | |
| Commercial assay or kit | Dynabeads M-270 Epoxy | Invitrogen | Cat# 14301 | |
| Commercial assay or kit | Pierce BCA Protein Assay Kit | Thermo Fisher Scientific | Cat# 23225 | |
| Commercial assay or kit | Mir-X miRNA First Strand Synthesis Kit | Takara | Cat# 638313 | |
| Commercial assay or kit | NEBNext Ultra Directional RNA Library Prep Kit | Illumina | Cat# E7420S | |
| Commercial assay or kit | NEBNext Multiplex Small RNA Library Prep Set | Illumina | Cat# **E7300S** | |
| Cell line (*M. musculus*) | ES-E14TG2a mESC | ATCC | CRL-1821 | |
| Cell line (*M. musculus*) | FLAG-Trim71 mESC | this paper | | |
| Cell line (*M. musculus*) | FLAG-Trim71 CLIPΔ mESC | this paper | | |
| Cell line (*M. musculus*) | FLAG-Trim71 *Ago2*Δ mESC | this paper | | |

*Continued*

| Reagent type (species) or resource | Designation | Source or reference | Identifiers | Additional information |
|---|---|---|---|---|
| Cell line (*M. musculus*) | FLAG-Trim71 *Dgcr8*Δ mESC | this paper | | |
| Cell line (*M. musculus*) | FLAG-Trim71 *Dicer*Δ mESC | this paper | | |
| Cell line (*M. musculus*) | FLAG-Trim71 CLIPΔ *Ago2*Δ mESC | this paper | | |
| Cell line (*M. musculus*) | FLAG-Trim71 CLIPΔ *Dgcr8*Δ mESC | this paper | | |
| Cell line (*M. musculus*) | FLAG-Trim71 CLIPΔ *Dicer*Δ mESC | this paper | | |
| Cell line (*M. musculus*) | FLAG-Ago2 mESC | this paper | | |
| Cell line (*M. musculus*) | FLAG-Trim71Δ mESC | this paper | | |
| Cell line (*M. musculus*) | FLAG-Trim71Δ CLIPΔ mESC | this paper | | |
| Recombinant DNA reagent | PiggyBac-based dox-inducible expression vector | this paper | pWH406 | |
| Recombinant DNA reagent | Inducible mouse FLAG-Trim71 expressing vector | this paper | pWH826 | |
| Recombinant DNA reagent | Inducible mouse FLAG-Trim71-C12AC15A expressing vector | this paper | pWH831 | |
| Recombinant DNA reagent | Inducible mouse FLAG-Trim71-R738A expressing vector | this paper | pWH840 | |
| Recombinant DNA reagent | Inducible mouse Ago2 expressing vector | this paper | pWH1070 | |
| Recombinant DNA reagent | Inducible GFP expressing vector | this paper | pWH1055 | |
| Recombinant DNA reagent | Inducible mouse V5-Lin28A expressing vector | this paper | pWH1081 | |
| Recombinant DNA reagent | Inducible mouse V5- Lin28B expressing vector | this paper | pWH1082 | |
| Recombinant DNA reagent | sgRNA and Cas9 expressing vector (pX458) pWH464 | Addgene | Cat# 48138 | |
| Recombinant DNA reagent | Super PiggyBac Transposase expressing vector (pWH252) | System Biosciences | Cat# PB210PA-1 | |
| Recombinant DNA reagent | The Luciferase reporter for measuring miR-293 activity | this paper | pWH854 | |
| Recombinant DNA reagent | FLuc-5BoxB reporter | PMID:28635594 | pWH290 | |
| Recombinant DNA reagent | The control reporter for the FLuc-5BoxB | PMID:28635594 | pWH291 | |
| Recombinant DNA reagent | lambdaN-GFP expressing plasmid | PMID:28635594 | pWH294 | |
| Recombinant DNA reagent | lambdaN-Trim71 expressing plasmid | this paper | pWH815 | |
| Recombinant DNA reagent | HCV-IRES bicistronic reporter | PMID:28635594 | pWH530 | |

*Continued on next page*

*Continued*

| Reagent type (species) or resource | Designation | Source or reference | Identifiers | Additional information |
|---|---|---|---|---|
| Recombinant DNA reagent | CrPV-IRES bicistronic reporter | PMID:28635594 | pWH531 | |
| Recombinant DNA reagent | FLuc-5BoxB-Malat1 reporter | PMID:28635594 | pWH569 | |
| Recombinant DNA reagent | FLuc-Malat1 reporter | PMID:28635594 | pWH570 | |
| Software, algorithm | FastQC v0.11.4 | Andrews S. 2010 | | https://www.bioinformatics.babraham.ac.uk/projects/download.html |
| Software, algorithm | Bowtie v1.1.2 | PMID:19261174 | | http://bowtie-bio.sourceforge.net/index.shtml |
| Software, algorithm | STAR v2.5.0 | PMID:23104886 | | https://github.com/alexdobin/STAR; *Lorenz et al., 2011* |
| Software, algorithm | Piranha v1.2.1 | PMID:23024010 | | http://smithlabresearch.org/software/piranha/ |
| Software, algorithm | iCount v2.0.1 | *Lovci et al., 2013* | | https://icount.readthedocs.io/en/latest/ |
| Software, algorithm | CLIPper v1.1 | *Lovci et al., 2013* | | https://github.com/YeoLab/clipper/wiki/CLIPper-Home; *Lovci et al., 2013* |
| Software, algorithm | CTK package v1.0.9 | PMID:27797762 | | https://zhanglab.c2b2.columbia.edu/index.php/CTK_Documentation |
| Software, algorithm | BEDtools v2.25.0 | PMID:20110278 | | https://bedtools.readthedocs.io/en/latest/ |
| Software, algorithm | SAMtools v0.1.19 | PMID:19505943 | | http://samtools.sourceforge.net/ |
| Software, algorithm | RNAfold v2.1.5 | PMID:22115189 | | https://www.tbi.univie.ac.at/RNA/ RNAfold.1.html |
| Software, algorithm | WebLogo v3.6.0 | PMID:15173120 | | http://weblogo.threeplusone.com/ |
| Software, algorithm | HISAT2 v2.1.0 | PMID:31375807 | | https://daehwankimlab.github.io/hisat2/ |
| Software, algorithm | HTSeq v0.11.1 | PMID:25260700 | | https://htseq.readthedocs.io/en/release_0.11.1 |
| Software, algorithm | R package EdgeR v3.26.8 | PMID:19910308 | | https://bioconductor.org/packages/release/bioc/html/edgeR.html |
| Software, algorithm | TargetScan v7.2 | PMID:26267216 | | http://www.targetscan.org/vert_72/ |

All the antibodies, plasmids, and oligonucleotides used in this study are listed in *Supplementary file 3*.

## mESC culture

All the mESCs described in this study are derived from ES-E14TG2a (ATCC CRL-1821). All the ES-E14TG2a derived mESCs used in this study were generated through CRISPR/Cas9-mediated genome editing, and their genotypes were confirmed by both PCR and western blot analysis. All the mESCs used in this study were cultured on 0.5% gelatin-coated tissue culture plates in either the 15% FBS + Lif (leukemia inhibitory factor) (medium DMEM/F-12 supplemented with 15% FBS, 2 mM L-glutamine, 0.1 mM MEM NEAA, 1% penicillin–streptomycin, 0.1 mM β-mercaptoethanol, and 1000 U/mL mLIF) or the 2i + Lif medium (DMEM/F-12, 2% FBS, 2 mM L-glutamine, 0.1 mM MEM NEAA, 1% penicillin–streptomycin, 0.1 mM β-mercaptoethanol and 1000 U/mL mLIF, 1 × N2N27, 3 μM CHIR99021 and 1 μM PD0325901). All the cells were grown in tissue culture incubators with temperature at 37°C and 5% $CO_2$.

## CRISPR/Cas9-mediated genome editing in mESCs

To generate the FLAG-Trim71 mESCs, 2 µg of pWH464 (pSpCas9(BB)−2A-GFP (pX458)) expressing the targeting sgRNA (oWH3373) and 1 µg of donor oligo oWH3375 was co-transfected into $1 \times 10^5$ E14 mESCs via the Fugene6 transfection reagents. To generate target gene knockout mESCs, 2 µg of pWH464 expressing a pair of sgRNAs target a coding region of the target gene was transfected into the mESCs. 24 hr post-transfection, top 10% GFP-positive cells were sorted into 96-well plates, with a single cell sorted into each well. After 7–14 days incubation, the correct mESC clones were screened and identified through genotyping PCR followed by western blot analysis.

## Trim71 CLIP-seq

The Trim71 CLIP-seq was performed using the previously established HITS-CLIP protocol (*Darnell, 2010*) with the following modifications. The FLAG-Trim71 mESCs were cross-linked by 0.4J UV254nm. The cell lysate was treated by RNase1 (Ambion) at 40 U/ml for 5 min at 37°C, and then 250 U/ml SUPERaseIn RNase Inhibitor (Ambion) was added to the cell lysate to inactivate the RNase1. 100 µl anti-FLAG M2-coupled Dynabeads M-270 (Invitrogen, Cat# 14301) per 10 mg cell lysate was used for the FLAG-Trim71 IP. Then 2.5% Input and IP samples were resolved on a 4–12% NuPage gel followed by transfer to a nitrocellulose membrane. Trim71–RNA complexes and size-matched input (*Van Nostrand et al., 2016*) were cut off from the membrane and were subject to RNA isolation and library preparation. The CLIP-seq libraries were sequenced on a HiSeq4000.

## qRT-PCR

For mRNA quantification, reverse transcription was performed on total RNA using random hexmers and Superscript2 reverse transcriptase (Thermo Fisher Scientific). miRNA and pre-miRNA quantification was performed in accordance with the protocols described previously (*Wan et al., 2010*) or using the Takara's Mir-X miRNA quantification method. qPCR was performed using the SsoAdvanced Universal SYBR Green Supermix (Bio-Rad) on a CFX Connect Real-Time PCR Detection System (Bio-Rad).

## Western blot

Cells were lysed in the RIPA buffer (10 mM Tris-HCl pH 8.0, 1 mM EDTA, 0.5 mM EGTA, 140 mM NaCl, 1% Triton X-100, 0.1% sodium deoxycholate, 0.1% SDS, and protease inhibitor cocktail). The cell lysate protein concentration was determined using a BCA assay kit from Thermo Fisher Scientific (Cat# 23225). Equal amount of cell lysate from the samples were resolved on SDS-PAGE gels and then transferred to PVDF membranes. The membranes were blocked with 3% non-fat dry milk in the TBS-T buffer for 1 hr at room temperature and then incubated with the indicated primary antibody overnight at 4°C. After washing, the appropriate horseradish peroxidase-conjugated secondary antibodies were applied for 1 hr at room temperature. Then the membranes were washed three times in the TBS-T buffer. The signals on the membranes were generated by the Clarity Western ECL substrate (Bio-Rad, Cat# 1705061), followed by X-ray film exposure. The exposed films were developed by an automatic Kodak film processor.

## Polysome analysis

Polysome analysis was performed using the protocol described previously (*Zhang et al., 2017*). Briefly, mESCs were lysed in the polysome lysis buffer (10 mM Tris-HCl pH 7.4, 12 mM MgCl$_2$, 100 mM KCl, 1% Tween-20, and 100 mg/ml cycloheximide). Then 5 OD260 cell lysate was loaded onto a 5–50% (w/v) linear sucrose-density gradient, followed by centrifugation at 39,000 rpm in a Beckman SW-41Ti rotor for 2 hr at 4°C. The gradient was fractionated using a Gradient Station (BioComp) coupled with an ultraviolet 254 nm detector (Bio-Rad EM-1).

## Colony formation assay for mESCs

500 mESCs/well were cultured in either the 15%FBS + Lif medium or the 2i + Lif medium in each well of a gelatinized 12-well plate for 7 days. The resultant colonies were then fixed and stained using an Alkaline Phosphatase Assay Kit (System Biosciences, Cat# AP100R-1). The morphology and AP intensity of the colonies were evaluated manually under an Olympus CK2 microscope. Each time 100–200 colonies from each type of mESCs were evaluated.

## Exit pluripotency assay for mESCs

ESCs were plated at a density of 1000 cells/well in a gelatinized 6-well plate in 2 ml of the differentiation medium (DMEM/F-12 supplemented with 15%FBS, 2 mM L-glutamine, 0.1 mM MEM NEAA, 1% penicillin–streptomycin, 0.1 mM β-mercaptoethanol) for 2 days. Then the medium was replaced with 3 ml of fresh 2i+Lif medium for another 5 days. Colonies were stained for alkaline phosphatase, and differentiation status was evaluated by the morphology and AP intensity.

## mESC differentiation

For EB formation, three million mESCs were cultured in 10 ml differentiation medium (DMEM/F-12 supplemented with 15% FBS, 2 mM L-glutamine, 0.1 mM MEM NEAA, 1% penicillin–streptomycin) in a 10 cm non-treated bacterial petri dish. The dish was placed on a horizontal rotator with a rotating speed of 30 rpm in a tissue culture incubator with temperature at 37°C and 5% $CO_2$. The medium was changed every other day, and the resulting EBs were harvested at the indicated time points. For monolayer differentiation, two million mESCs were cultured in 10 ml differentiation medium on a gelatinized 9 cm tissue culture dish. The dish was placed in a tissue culture incubator with temperature at 37°C and 5% $CO_2$. The resultant cells were harvested at the indicated time points.

Cell Proliferation Assay mESCs were plated in gelatin coated 24-well plates at 20,000 cells per well, and their proliferation was assessed every day for 3 days using the CellTiter 96 AQueous One Solution Reagent (Promega, Cat# G3582). The measurement was performed in accordance with the protocol provided by the kit.

## RNA-seq and small-RNA-seq analysis

The reads from RNA-seq and small-RNA-seq were mapped to the mm10 genome by using HISAT2 (v2.1.0). The mapping results were converted into bam format by using SAMtools. The read count for the longest transcript of each gene was calculated by using HTSeq (v0.11.2) and was then converted into TPM value. The genes with TPM $\geq$ 1 in RNA-seq and miRNAs with TPM $\geq$ 100 in small-RNA-seq were kept for further analysis. The Negative Binomial Generalized Linear Models with Quasi-Likelihood Tests function in EdgeR was used for differential expression analysis. The p-value cut off for differential expressed miRNAs was set to 0.05. The targets of non-differential expressed miRNAs are obtained from the 'predicted conserved targets' table v7.2 generated by the TargetScan (*Agarwal et al., 2015*). Mann–Whitney U-test was used to evaluate the probability that the microRNA targets and non-microRNA targets have the same distribution of fold change in expression level between wild type and KO data sets. The scatter plot and the cumulative plots were generated by using python package matplotlib. The correlation matrix of miRNA data sets was visualized by using the R package corrplot and psych.

## CLIP-seq peak calling

The CLIP-seq reads were quality-checked by using FastQC. The reads from rRNA, tRNA, and mitochondrial DNA sequences were removed from data sets by using Bowtie. The resulting reads were mapped to the mm10 genome by using STAR (v2.5.0) with the parameters suggested in a previous study (*Van Nostrand et al., 2016*). CLIPper (*Lovci et al., 2013*) was used to call peaks for the two replicates over the input control data sets. The complement set of peak calling results was kept as the background for the motif study. The peaks were annotated to the mm10 RefSeq mRNAs by bed2annotation tool in the CTK package. To determine secondary structure motifs in the CLIP-seq peak regions, each peak was extended from the peak center to a 50 nt binding bins. The background regions are divided into bins of 50 nt. The RNA sequences of both the binding and non-binding bins were obtained by using the getfasta function in the BEDtools with the parameter '-s'. RNAfold (2.1.5) (*Lorenz et al., 2011*) was used to predict the minimum free energy (MFE) secondary structures for both binding and non-binding bins. The enrichment of each 11-mer secondary structure substring was calculated from the dot-bracket encoded MFE secondary structures of both binding and non-binding bins. The 11-mer secondary structure substrings with counts <10 or do not contain any hairpin loop were discarded. The secondary structure motif logo was generated from the secondary structure substrings with enrichment $\geq$ 1 by using WebLogo (v3.6.0).

The CLIP-seq, RNA-seq, and small-RNA-seq data sets generated during this study are available at GEO: GSE138284.

## Acknowledgements

We thank Drs. Juan R Alvarez-Dominguez and Lei Sun for critical comments, and Dr. Jianfu Chen for the FLAG-Trim71 plasmid. This work was supported by NIH grants (R01HL141112, R01GM136869, and R21AI146431) and Mayo Foundation for Medical Education and Research.

## Additional information

### Funding

| Funder | Grant reference number | Author |
|---|---|---|
| National Heart, Lung, and Blood Institute | R01HL141112 | Qiuying Liu<br>Mariah K Novak<br>Wenqian Hu |
| National Institute of General Medical Sciences | R01GM136869 | Qiuying Liu<br>Mariah K Novak<br>Wenqian Hu |
| National Institute of Allergy and Infectious Diseases | R21AI146431 | Xiaoli Chen<br>Shaojie Zhang<br>Wenqian Hu |

The funders had no role in study design, data collection and interpretation, or the decision to submit the work for publication.

### Author contributions

Qiuying Liu, Data curation, Formal analysis, Investigation, Methodology, Writing - original draft; Xiaoli Chen, Software, Formal analysis; Mariah K Novak, Formal analysis, Investigation, Writing - original draft; Shaojie Zhang, Supervision; Wenqian Hu, Conceptualization, Data curation, Formal analysis, Supervision, Funding acquisition, Investigation, Methodology, Writing - original draft, Project administration, Writing - review and editing

### Author ORCIDs

Qiuying Liu (iD) https://orcid.org/0000-0002-1474-4487
Mariah K Novak (iD) http://orcid.org/0000-0003-1547-4738
Shaojie Zhang (iD) http://orcid.org/0000-0002-4051-5549
Wenqian Hu (iD) https://orcid.org/0000-0003-3577-3604

### Decision letter and Author response

Decision letter https://doi.org/10.7554/eLife.66288.sa1
Author response https://doi.org/10.7554/eLife.66288.sa2

## Additional files

### Supplementary files

• Supplementary file 1. Trim71-binding regions in its target mRNA identified by the CLIP-seq.

• Supplementary file 2. miRNAs detected in the WT and the CLIPΔ mESCs. The first tab lists the differentially expressed miRNAs, and the second tab lists the non-differentially expressed miRNAs. The expression level is indicated by reads per million (RPM).

• Supplementary file 3. Antibodies, plasmids, and oligonucleotides used in this study.

• Transparent reporting form

### Data availability

The CLIP-seq, RNA-seq, small-RNA-seq datasets generated during this study are available at GEO: GSE138284.

The following dataset was generated:

| Author(s) | Year | Dataset title | Dataset URL | Database and Identifier |
|---|---|---|---|---|
| Hu W, Liu Q, Zhang H, Chen X, Zhang S | 2020 | Studies on Trim71 in mouse embryonic stem cells | https://www.ncbi.nlm.nih.gov/geo/query/acc.cgi?acc=GSE138284 | NCBI Gene Expression Omnibus, GSE138284 |

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
