## [Decision Letter]

**Acceptance summary:**

The study represents a substantial amount of work consisting of gain- and loss-of-function approaches to investigate the molecular mechanisms by which the evolutionarily conserved RNA-binding protein Trim71 regulates mammalian stem cell pluripotency. The findings pave the way for understanding how an ancient microRNA pathway regulates stem cell functions.

**Decision letter after peer review:**

[Editors’ note: the authors submitted for reconsideration following the decision after peer review. What follows is the decision letter after the first round of review.]

Thank you for submitting your work entitled "Repressing Ago2 mRNA translation by Trim71 maintains pluripotency through inhibiting let-7 microRNAs" for consideration by *eLife*. Your article has been reviewed by a Senior Editor, a Reviewing Editor, and three reviewers. The reviewers have opted to remain anonymous.

Our decision has been reached after consultation between the reviewers. Based on these discussions and the individual reviews below, we regret to inform you that your work cannot be considered further for publication in *eLife*, at least in its current form.

There was significant enthusiasm for the work. However, it seems that considerable effort including additional experiments will be required to firm up the conclusions and make the paper suitable for *eLife*. We believe that required revisions cannot be completed within two months. Accordingly, we must reject the paper in its current form. Nevertheless, we encourage you to resubmit if and when you can address the majority of the reviewers' concerns.

Reviewer #1:

Liu et al., examine the mechanism by which Trim71 controls the activity of the let-7 microRNA (miRNA) family in stem cells. The work is based on previous studies, predominantly in *C. elegans*. In essence, the authors assert that Trim71 maintains pluripotency by repressing the translation of the Ago2 transcript, thereby impairing the activity of miRNAs, with repression of the let-7 family playing a key role in maintaining stemness. The key tool developed here is a FLAG-tagged Trim71, which is used to reveal the binding preferences for Trim71 (using CLIP-seq). The experiments are logical and solid, although it needs to be acknowledged that there is relatively little data, and no pivotal experiments that strongly support the model. The manuscript is well-written, and potentially suitable for publication in *eLife*, although the concerns raised below should be addressed prior to publication.

1) As cited in the manuscript, previous publications describe different mechanisms by which Trim71 functions. The current manuscript implies a different mechanism, but much of the data is correlative – key experiments that robustly support the authors model are, largely, lacking. The work would be significantly more impactful if additional data could be added to support the model presented. Many, many such experiments are possible. For example, what would happen if Ago1, 3 or 4 were increased (akin to the experiments in which the authors manipulate Ago2); how does Trim71 control translation (no data is provided for this point)? Additional experiments are suggested below.

2) The authors suggest that the number of Trim71 binding sequences govern whether translation of stability of the transcript is controlled – is there precedence for this model? How do the authors envision this working?

3) It would be useful to add a simple experiment showing that the mouse Ago2 transcript is also subject to translational regulation by Trim71.

4) A rescue experiment in which Trim71 is tethered to the mutant Ago2 transcript could provide some convincing additional support for the model. Similarly, what happens when Trim71 is tethered to other transcripts?

5) Why does inhibition of Ago2 via Trim71 inhibit specifically the let-7 family? The authors provide data that suggests specificity, but they perform no experiments to define the mechanism. Given that previous publications assert mechanisms for Trim71 that are markedly different than that proposed here, it is appropriate for Liu et al. to provide data supporting a key aspect of their model.

6) A key part of the work relies upon showing that pri-let-7 transcripts are not changed in response to the Trim71 family. Similarly, the authors should investigate whether pre-let-7 transcripts are changing. The authors' model predicts that pre-let-7 will remain unchanged – confirming this would significantly strengthen the model. This experiment is important, as multiple other papers (cited in this manuscript) propose alternative mechanisms for Trim71 function.

Reviewer #2:

Liu et al., studied the mechanism by which Trim71 inhibits let-7 miRNA levels. Using CLIP, the authors identified genome-wide binding patterns of Trim71 in murine ESCs, and identified a binding peak in Ago2 3'UTR. Although knocking out Trim71 did not affect Ago2 protein level, the authors proposed that Trim71 inhibits Ago2 translation based on several pieces of evidence, including mutant ESCs (deltaCLIP) with deletion of the Trim71 binding region in Ago2 3'UTR, as well as polysome fractionation experiments. In the δ-CLIP mutant ESCs, Ago2 protein levels were increased by 2 fold. This 2 fold increase of Ago2 was associated with a preferential increase in let-7 miRNAs over other miRNAs, and the causality of this effect was demonstrated by ectopic expression of Ago2 in mESCs. The authors further demonstrated that the δ CLIP clones had reduced self-renewal and enhanced potential to differentiate, which were dependent on Ago2 on let-7.

Although the effect of Trim71 on ESC biology and the regulation of Trim71 by let-7 have been well reported before, the identification of specific regulation of Trim71 on Ago2, if proven solidly, is interesting. Additionally, a preferential regulation of let-7 by Ago2 levels is also interesting for the field. With these said, I do have some concerns about the data and have some suggestions to improve the manuscript.

1) The deltaCLIP clones are critical reagents in this study, yet the genetics are not well described. According to Figure 2A and Figure 2B, there seems to be only one band amplifiable by PCR in each of the deltaCLIP clones, and there seems to be only one allele in each of the clones. This is somewhat surprising given the use of CRISPR in generating such clones. One possibility is that there are identical deletions on the two copies of Ago2 gene. The second possibility is that one of the Ago2 alleles harbors a much larger deletion that evades detection by PCR and subsequent sequencing. So proving the genetics of the clones is critical for data interpretation.

2) The effect of preferential let-7 accumulation in the presence of increased Ago2 is not easily expected, given that there are multiple Ago family proteins expressed in mESCs and that single KO of Ago2 does not cause any major phenotypes in mESCs. I wonder whether the mechanism could be due to Ago2 processing pre-let-7 in a Dicer independent manner. This possibility is also suggested by the fact that the majority of deregulated let-7 miRNAs are 5p-miRNAs (Figure 5A). This can be tested by examining let-7 vs other miRNAs in the dicerKO/deltaCLIP line. Additionally, expressing a slider activity dead Ago2 in Ago2KO/detaCLIP line can also give some clues.

3) The self-renewal assay was based on the number of AP+ colonies. This is a useful assay, but it does not exclude the possibility of colony size being different. I suggest the authors also run a competition assay-for example, labeling the deltaCLIP clone with a fluorescent protein, and then mix it with the WT clone in 1:1 ratio, and then follow the ratio with FACS each time the cells are passaged.

4) Figure 7F is an important figure to show the direct involvement of let-7 in deltaCLIP clones. However, with control LNA, the colony in deltaCLIP is about 25% of WT control, yet in most other figures, e.g Figure 4A, Figure 7C, the difference is only about 50%. Could the authors comment on whether this is due to side effects of LNA, and how reproducible the data in Figure 7F are?

Reviewer #3:

The study by Liu et al., is intriguing and is likely to be controversial, given its argument for Ago2 as an important regulatory target of Trim71 in contrast to other published work in the field. Despite this potential controversy, the study provides multiple lines of convincing evidence and experimental paradigms to support their assertions, including a multitude of loss- and gain-of-function studies. The Trim71-let7 axis is a fascinating molecular pathway that is conserved across phylogeny, yet its cellular and molecular functions and underlying mechanisms are poorly understood, especially in mammals. The study represents an impressive amount of work that should stimulate much-needed discussion in the field and pave the way for future lines of investigation to better understand how this evolutionarily conserved microRNA pathway controls organogenesis and development.

---

## [Author Response]

[Editors’ note: the authors resubmitted a revised version of the paper for consideration. What follows is the authors’ response to the first round of review.]

Reviewer #1:Liu et al., examine the mechanism by which Trim71 controls the activity of the let-7 microRNA (miRNA) family in stem cells. The work is based on previous studies, predominantly in *C. elegans*. In essence, the authors assert that Trim71 maintains pluripotency by repressing the translation of the Ago2 transcript, thereby impairing the activity of miRNAs, with repression of the let-7 family playing a key role in maintaining stemness. The key tool developed here is a FLAG-tagged Trim71, which is used to reveal the binding preferences for Trim71 (using CLIP-seq). The experiments are logical and solid, although it needs to be acknowledged that there is relatively little data, and no pivotal experiments that strongly support the model. The manuscript is well-written, and potentially suitable for publication in eLife, although the concerns raised below should be addressed prior to publication.

We appreciate that this reviewer believes that “the experiments are logical and solid” and “this manuscript is well-written”. We provide our response to his/her comments below.

1) As cited in the manuscript, previous publications describe different mechanisms by which Trim71 functions. The current manuscript implies a different mechanism, but much of the data is correlative – key experiments that robustly support the authors model are, largely, lacking. The work would be significantly more impactful if additional data could be added to support the model presented. Many, many such experiments are possible. For example, what would happen if Ago1, 3 or 4 were increased (akin to the experiments in which the authors manipulate Ago2); how does Trim71 control translation (no data is provided for this point)? Additional experiments are suggested below.

We performed the experiment suggested by this reviewer (Figure 5—figure supplement 1). Specifically, we chose to increase Ago1 level, because in mESCs Ago1 and Ago2 are expressed, while Ago3 and Ago4 are not (Figure 5—figure supplement 1B). We found that similar to the results of Ago2 (Figure 5), when Ago1 was induced, there was a specific post-transcriptional increase of mature let-7 miRNAs (Figure 5—figure supplement 1D,E). The result further supports that the Ago proteins are the limiting factors in let-7 miRNPs.

2) The authors suggest that the number of Trim71 binding sequences govern whether translation of stability of the transcript is controlled – is there precedence for this model? How do the authors envision this working?

We proposed this model to reconcile our finding that Trim71 represses Ago2 mRNA translation with the previous results that Trim71 promotes mRNA degradation. Since mRNA translation and mRNA degradation are highly intertwined with each other, we speculated that maybe the number of Trim71-binding sites determine whether Trim71 represses translation or destabilizes mRNA. We realized that this model is highly speculative. Thus, in the revised manuscript, we deleted this speculation and revised the Discussion section accordingly.

3) It would be useful to add a simple experiment showing that the mouse Ago2 transcript is also subject to translational regulation by Trim71.

In the manuscript, we used multiple lines of evidence to show that mouse Ago2 mRNA is translationally repressed by Trim71 in mESCs:

a) when Trim71-binding site was disrupted in the 3’UTR of Ago2 mRNA, Ago2 level increased ~2-fold, while mRNA level was not altered (Figure 2A-C).

b) Ago2 mRNA showed increased polysome association in the CLIPD mESCs (Figure 2D,E).

c) Overexpression of Trim71 did not change Ago2 mRNA level, but decreased Ago2 protein level and Ago2 mRNA association with polysomes (Figure 2G,H).

d) Using the new results from the tethering experiment (Figure 7), we also found that Trim71 represses mRNA translation whether tethered to a reporter mRNA.

4) A rescue experiment in which Trim71 is tethered to the mutant Ago2 transcript could provide some convincing additional support for the model. Similarly, what happens when Trim71 is tethered to other transcripts?

We performed similar experiments in the revised manuscript as the reviewer suggested (Figure 7). Specifically, we used tethering assays to show that when tethered to a reporter mRNA, Trim71 can represses the translation (Figure 7). Mover, using IRES-containing bi-cistronic reporters, we took a step further by determining that Trim71 represses mRNA translation at post-initiation step(s) in mESCs.

5) Why does inhibition of Ago2 via Trim71 inhibit specifically the let-7 family? The authors provide data that suggests specificity, but they perform no experiments to define the mechanism. Given that previous publications assert mechanisms for Trim71 that are markedly different than that proposed here, it is appropriate for Liu et al. to provide data supporting a key aspect of their model.

We agree with the reviewer that it is important to show how elevated Ago2 specifically increase let-7 miRNAs. Using new data presented in Figure 5—figure supplement 2, we addressed this point. Specifically, we found that the increased let-7 miRNAs resulting from elevated Ago2 in mESCs are bound and stabilized by Ago2. Moreover, we added text (Results) explaining that the pro-differentiation let-7 miRNAs are unique in mESCs because although genes encoding let-7 miRNAs are actively transcribed, the let-7 miRNAs levels are low, indicating post-transcriptional regulations of let-7 miRNAs. Previous studies characterized one such regulation at the pre-miRNA processing level mediated by Lin28a. Here we showed that, at the mature miRNA level, let-7 miRNA level and stability are regulated by Ago2 availability in mESCs.

6) A key part of the work relies upon showing that pri-let-7 transcripts are not changed in response to the Trim71 family. Similarly, the authors should investigate whether pre-let-7 transcripts are changing. The authors' model predicts that pre-let-7 will remain unchanged – confirming this would significantly strengthen the model. This experiment is important, as multiple other papers (cited in this manuscript) propose alternative mechanisms for Trim71 function.

We agree with the reviewer that this is an important experiment. In the revised manuscript, we analyzed the let-7 pre-miRNAs levels in the CLIPΔ mESCs (Figure 4C). We found that although pre-let-7 miRNAs were elevated in the CLIPD mESCs (Figure 4C), the 2-3-fold increase of let-7 pre-miRNAs was not at the same magnitude as the increased mature let-7 miRNAs (Figure 4C versus Figure 4B). This modest increase of let-7 pre-miRNA is consistent with the observation that Lin28a, a conserved let-7 miRNA target that promotes let-7 pre-miRNA degradation, decreased ~2 fold in the CLIPΔ mESCs (Figure 4D). This finding also revealed the regulations on mature let-7 miRNAs is tightly intertwined: as Ago2 level elevation results in an increase of mature let-7, which leads to downregulation of Lin28a, a let-7 target that promotes degradation of let-7 pre-miRNAs. This downregulation of Lin28a further reinforces let-7 expression. In addition to adding the new results on let-7 pre-miRNA levels, we also revised the Discussion section on the tightly intertwined regulation of let-7 miRNAs.

Reviewer #2:Liu et al., studied the mechanism by which Trim71 inhibits let-7 miRNA levels. Using CLIP, the authors identified genome-wide binding patterns of Trim71 in murine ESCs, and identified a binding peak in Ago2 3'UTR. Although knocking out Trim71 did not affect Ago2 protein level, the authors proposed that Trim71 inhibits Ago2 translation based on several pieces of evidence, including mutant ESCs (deltaCLIP) with deletion of the Trim71 binding region in Ago2 3'UTR, as well as polysome fractionation experiments. In the δ-CLIP mutant ESCs, Ago2 protein levels were increased by 2 fold. This 2 fold increase of Ago2 was associated with a preferential increase in let-7 miRNAs over other miRNAs, and the causality of this effect was demonstrated by ectopic expression of Ago2 in mESCs. The authors further demonstrated that the δ CLIP clones had reduced self-renewal and enhanced potential to differentiate, which were dependent on Ago2 on let-7.Although the effect of Trim71 on ESC biology and the regulation of Trim71 by let-7 have been well reported before, the identification of specific regulation of Trim71 on Ago2, if proven solidly, is interesting. Additionally, a preferential regulation of let-7 by Ago2 levels is also interesting for the field. With these said, I do have some concerns about the data and have some suggestions to improve the manuscript.

We appreciate the reviewer’s comment that our finding is interesting for the field. We provide our response to his/her comments below.

1) The deltaCLIP clones are critical reagents in this study, yet the genetics are not well described. According to Figure 2A and Figure 2B, there seems to be only one band amplifiable by PCR in each of the deltaCLIP clones, and there seems to be only one allele in each of the clones. This is somewhat surprising given the use of CRISPR in generating such clones. One possibility is that there are identical deletions on the two copies of Ago2 gene. The second possibility is that one of the Ago2 alleles harbors a much larger deletion that evades detection by PCR and subsequent sequencing. So proving the genetics of the clones is critical for data interpretation.

We agree with the reviewer that providing the genetics of the CLIPD mESC clones is critical for the study. In the revised manuscript, we provided the additional data to strengthen this point.

Specifically, first, we observed similar RNA-seq reads intensity and distribution across the whole *Ago2* 3’UTR except the deleted Trim71-binding region (CLIPΔ) among the WT and the two CLIPD clones (Figure 2—figure supplement 1C). This result indicated that there is no large DNA fragment deletion caused by the genome editing in the target region (Ago2 3’UTR).

Second, we showed that different from the results in the WT background, in the *Trim71*D genetic background, the CLIPD in the 3’UTR of *Ago2* mRNA did not alter Ago2 level (Figure 2—figure supplement 1F vs Figure 2B). This result indicated that the Trim71-binding site in the 3’UTR of *Ago2* mRNA does not regulate Ago2 mRNA translation *in cis* and is dependent on Trim71 to regulate Ago2 expression.

2) The effect of preferential let-7 accumulation in the presence of increased Ago2 is not easily expected, given that there are multiple Ago family proteins expressed in mESCs and that single KO of Ago2 does not cause any major phenotypes in mESCs. I wonder whether the mechanism could be due to Ago2 processing pre-let-7 in a Dicer independent manner. This possibility is also suggested by the fact that the majority of deregulated let-7 miRNAs are 5p-miRNAs (Figure 5A). This can be tested by examining let-7 vs other miRNAs in the dicerKO/deltaCLIP line. Additionally, expressing a slider activity dead Ago2 in Ago2KO/detaCLIP line can also give some clues.

We agree with the reviewer that it is important to test the possibility of Ago2-mediated processing of pre-miRNAs, which was also reported in the literature (e.g. Diederichs and Haber, 2007). Thus, we measured the let-7 pre-miRNAs in the CLIPΔ mESCs, where Ago2 level was elevated ~2-fold. We found that the pre-let-7 miRNAs were also in the CLIPD mESCs (Figure 4C), although the increased fold was not as high as those of mature let-7 miRNAs. Thus, this result argues against the possibility of Ago2mediated processing of pre-miRNAs.

The second point raised by the reviewer is why there is a preferential increase of let-7 mature miRNAs when Ago2 is increased. We addressed this using new data presented in Figure 5—figure supplement 2. We found that the increased let-7 miRNAs resulting from elevated Ago2 in mESCs are bound and stabilized by Ago2. Moreover, we added text (Results) explaining that the pro-differentiation let-7 miRNAs are unique in mESCs because although genes encoding let-7 miRNAs are actively transcribed, the let-7 miRNAs levels are low, indicating post-transcriptional regulations of let-7 miRNAs. Previous studies characterized one such regulation at the pre-miRNA processing level mediated by Lin28a. Here we showed that, at the mature miRNA level, let-7 miRNA level and stability are regulated by Ago2 availability in mESCs.

3) The self-renewal assay was based on the number of AP+ colonies. This is a useful assay, but it does not exclude the possibility of colony size being different. I suggest the authors also run a competition assay-for example, labeling the deltaCLIP clone with a fluorescent protein, and then mix it with the WT clone in 1:1 ratio, and then follow the ratio with FACS each time the cells are passaged.

The assay described by the reviewer measures cell growth, while not stemness or self-renewal.

In measuring stemness or self-renewal, it is important to track the parental-progeny relationships of the cells. In the colony formation assay, all the cells in one colony are from a single parental mESC. By examining whether all these progeny cells in the colony maintains the same feature or not (e,g, whether all of them are AP+ or not), we can determine whether stemness/self-renewal is maintained during division.

In our colony formation assay, we didn’t notice any colony size difference between the

WT and the CLIPD mESCs. This is consistent with the results when we assayed for cell growth/proliferation: there was no difference between WT and the CLIPD mESCs in cell proliferation (Figure 2—figure supplement 1G).

4) Figure 7F is an important figure to show the direct involvement of let-7 in deltaCLIP clones. However, with control LNA, the colony in deltaCLIP is about 25% of WT control, yet in most other figures, e.g Figure 4A, Figure 7C, the difference is only about 50%. Could the authors comment on whether this is due to side effects of LNA, and how reproducible the data in Figure 7F are?

We thank the reviewer for pointing out this. This difference is due to the potential side effects of LNA and/or transfecting reagents. Also, we are confident about the results. Because we actually did the transfection experiments at two different dosage of LNA. As shown in Author response image 1, at low LNA dosage (100nM), the anti-let-7 LNA can increase the stemness from ~5% to ~10% (AP+ colonies); while at the high dosage (500nM), the anti-let-7 LNA can further increase the stemness to ~15% (AP+ colonies). Importantly, the control LNA showed no difference between the low dosage and the high dosage. Since the transfection can never be 100%, the dosage dependent effect strongly argues that the stemness defects in the CLIPD mESCs are dependent on the let-7 miRNAs. In the manuscript, due to space limitations, we only showed the results from the high dosage of LNA transfection.
